# Precision Nutrition and the Microbiome Part II: Potential Opportunities and Pathways to Commercialisation

**DOI:** 10.3390/nu11071468

**Published:** 2019-06-27

**Authors:** Susan Mills, Jonathan A. Lane, Graeme J. Smith, Keith A. Grimaldi, R. Paul Ross, Catherine Stanton

**Affiliations:** 1APC Microbiome Ireland, University College Cork, T12 K8AF Cork, Ireland; 2H&H Group, Technical Centre, Global Research and Technology Centre, P61 C996 Cork, Ireland; 3Eurogenetica Ltd., Burnham-on-Sea TA8 1HX, UK; 4APC Microbiome Ireland, Teagasc Food Research Centre, P61 C996 Fermoy, Cork, Ireland

**Keywords:** personalised nutrition, precision nutrition, probiotics, prebiotics, gut microbiome, immunity, metabolic disease, gut, genetics

## Abstract

Modulation of the human gut microbiota through probiotics, prebiotics and dietary fibre are recognised strategies to improve health and prevent disease. Yet we are only beginning to understand the impact of these interventions on the gut microbiota and the physiological consequences for the human host, thus forging the way towards evidence-based scientific validation. However, in many studies a percentage of participants can be defined as ‘non-responders’ and scientists are beginning to unravel what differentiates these from ‘responders;’ and it is now clear that an individual’s baseline microbiota can influence an individual’s response. Thus, microbiome composition can potentially serve as a biomarker to predict responsiveness to interventions, diets and dietary components enabling greater opportunities for its use towards disease prevention and health promotion. In Part I of this two-part review, we reviewed the current state of the science in terms of the gut microbiota and the role of diet and dietary components in shaping it and subsequent consequences for human health. In Part II, we examine the efficacy of gut-microbiota modulating therapies at different life stages and their potential to aid in the management of undernutrition and overnutrition. Given the significance of an individual’s gut microbiota, we investigate the feasibility of microbiome testing and we discuss guidelines for evaluating the scientific validity of evidence for providing personalised microbiome-based dietary advice. Overall, this review highlights the potential value of the microbiome to prevent disease and maintain or promote health and in doing so, paves the pathway towards commercialisation.

## 1. Introduction

The gut microbiota is an integral component of the human body, and such is its contribution to human physiology that it has been deemed an organ in itself. With a genetic coding capacity that exceeds its human host by ≥100-fold [1], the gut microbiota executes essential functions that the body itself is incapable of performing. It promotes gut maturation, educates the immune system, provides protection against viral and bacterial pathogens, influences brain activities and bodily metabolism. In Part I of this two-part review [2], we provided an overview of its development from birth to old age and detailed how it impacts host health through multiple mechanisms.

Importantly, several factors influence its composition and activities, one of which is host genetics, a factor which is beyond our control, while another significant contributor to its form and function is diet, an element which we can control. Indeed, humans not only feed themselves but also feed their gut microbiota. These two factors alone (host genetics and diet) largely account for the huge variability in microbiome composition and functionality which exists among individuals. Indeed, such is the inter-individual variability that scientists still grapple with what constitutes a “healthy” microbiota. One feature of a poorly functioning microbiota that is incapable of serving its host to its full potential is low microbial diversity. Indeed, in Part I of this review we discussed the implications of low microbial diversity in terms of infection and inflammation, the latter of which is associated with several non-communicable diseases in its chronic form including cardiovascular diseases, diabetes, allergies and arthritis as examples. Improving microbial diversity can be achieved through healthy eating and consuming the recommended daily intake for fibre (25 g/day for women and 38 g/day for men [3]). In Part I of this review we discussed the role of diet in shaping the microbiome with a particular focus on the Mediterranean diet. Long-term consumption of this diet not only improved the microbial profile and actions of the gut inhabitants in obese men but also generated physiological improvements in terms of metabolism [4,5]. In terms of feeding our gut microbiota “long-term” healthy dietary patterns appear to be the key since short term dietary interventions of this nature have minimal, if any, impact on microbiota diversity levels [6,7].

Interventions involving probiotics, prebiotics, synbiotics and dietary fibre also offer opportunities to “fertilize” our microbiota. Probiotics are defined as ‘live microorganisms, which when administered in adequate numbers confer a health benefit on the host’ [8]. The following genera represent the most commonly used probiotics for which health claims have been demonstrated, and within these, the benefits tend to be strain specific: *Bifidobacterium*, *Lactobacillus*, *Saccharomyces*, *Streptococcus*, *Enterococcus*, *Leuconostoc*, *Pediococcus*, *Escherichia coli* and *Bacillus* [9]. However, the prerequisite for ‘live microorganisms’ is subject to some debate, given that a pasteurised derivative of a beneficial strain exhibited enhanced effects in obese and diabetic mice [10]. The prebiotic definition has been recently updated/broadened to “a substrate that is selectively utilized by host microorganisms conferring a health benefit” by the International Scientific Association for Probiotics and Prebiotics [11]. By modulating the intestinal microbiota with a high or low level of specificity and increasing the abundance of beneficial bacteria, prebiotics can improve host metabolic and physiological parameters. Synbiotics describe the combination of probiotics and prebiotics which act synergistically. Dietary fibre has been defined as “the edible part of plants or their extracts, or analogous carbohydrates, that are resistant to digestion in the human small intestine, and undergoes complete or partial fermentation in the large intestine” [12], or more simply as “any dietary component that reaches the colon without being absorbed in a healthy gut” [13].

In this review, we examine initially the consequences of different life stages or situations on the gut microbiota of humans and examine the efficacy of probiotics and prebiotics with a focus on gut microbiota modulation and/or improvement of symptom(s). We then investigate the potential of probiotics, prebiotics and dietary fibre to aid in the management of two forms of malnutrition which are prevalent in both developed and developing countries, namely, overnutrition and undernutrition, reporting changes conveyed to the gut microbiota and hence host physiology based on data from human studies. However, it is becoming increasingly clear that an individual’s baseline microbiota and genetic make-up can influence the efficacy of such interventions and scientists are beginning to unravel the discrepancies which exist between human ‘responders’ and ‘non-responders.’ This is perhaps one of the core elements of precision nutrition through the microbiome whereby it can serve as a biomarker to predict responsiveness to dietary components and interventions. As an example, the gut microbiota of an individual can be used to predict postprandial glycemic responses (PPGRs) to food [14] enabling the design of a precision-tailored individualised diet that helps prevent the development of metabolic syndrome and its comorbidities, a study which is discussed in more detail in Section 5. This level of data paves the way for new opportunities in terms of interventions and microbiome testing at an individual level. Microbiome testing is currently available; thus, we discuss its feasibility at this moment in time and how it can be streamlined to generate more scientifically meaningful results. Finally, we propose guidelines for evaluating the scientific validity of evidence for providing personalised microbiome-based dietary advice.

## 2. Impact of Environment and Life Stage on Gut Microbiota and Health and Opportunities for Optimising Health through Diet, Probiotics and Prebiotics

As science continues to delineate the composition and functionality of life stage-specific gut microbiota and deviations from what is considered “normal” or “healthy,” opportunities arise for dietary and therapeutic interventions which can beneficially modulate the microbiota and result in translational benefits to host physiology and overall health. In this section, we consider different life stages/situations and the impact of each on the gut microbiota including pregnancy, infancy and the elderly, especially focusing on those in long-stay care facilities, physical activity, and times of psychological stress. Dietary recommendations exist for these particular life junctures, but we also summarise a number of studies which have investigated the potential of probiotics and prebiotics to beneficially influence the gut microbiota and ultimately human health.

### 2.1. Pregnancy

The female body undergoes several changes during pregnancy including an increase in body fat in early pregnancy which is followed by a decrease in insulin sensitivity later on [15]. The change in insulin sensitivity has been linked to immunity changes which are proposed to induce metabolic inflammation that is normally associated with obesity [16]. However, during pregnancy these changes support the growth of the foetus and prepare the mother’s body for lactation [17,18,19]. Specific nutritional recommendations exist for pregnancy, but these can differ depending on eating tradition and nutritional status of the population [20]. However, the gut microbiota of the pregnant mother has received increasing attention given that it can influence the health of both mother and child.

In a study involving 91 pregnant mothers of varying body mass index (BMI) and gestational diabetes, Koren et al. [21] reported that the gut microbiota changes dramatically from the first trimester (T1) to the third trimester (T3) even though the diets and energy intake of participants did not change between sampling times. From T1 to T3, Proteobacteria significantly increased in 69.5% of women and Actinobacteria increased in 57% of women. As women progressed from T1 to T3, the number of operational taxonomic units (OTUs) became significantly reduced and T1 samples had greater within-sample alpha phylogenetic diversity than T3 samples irrespective of pre-pregnancy BMI and health status. It has been suggested that the reduced alpha diversity in T3 may not be due to loss of species but rather lower relative abundance levels below the sequencing level of detection [22]. The over-represented OTUs in T1 mainly belonged to the Clostridiales order of the Firmicutes and included butyrate producers such as *Facalibacterium* and *Eubacterium* [21]. Members of the *Enterobacteriaceae* family and the *Streptococcus* genus were over-represented in T3 samples. It is speculated that the increase in butyrate-producing microorganisms in T1 could increase immunoregulatory T regulatory (T-reg) cells which may be involved in reducing maternal rejection of the foetal allograft [22]. Interestingly, no correlations were found between the specific OTU abundance and the use of antibiotics, probiotics, diet, previous pregnancies or health markers [21]. The results revealed that T1 microbial diversity is similar to the microbial diversity observed in non-pregnant women while T3 microbial diversity is aberrant and persists for one month postpartum. In T3 and just before transmission of the microbiota to the newborn, each mother has a “purely personal” microbiota which is suggested to have been selected at the level of each host lineage to ensure maximum development of the developing foetus and newborn [22]. Transferring T3 microbiota to germ-free mice resulted in increased adiposity and reduced insulin sensitivity compared to T1 microbiota [21]. The study indicates that the microbial changes which occur during pregnancy influence host metabolism and are beneficial for that stage in life. It is suggested that such changes are driven by the immunological and hormonal changes which occur during pregnancy [22]. However, a follow-up study conducted in 2015 investigating temporal and spatial variation of the human microbiota at four body sites (distal gut, vagina, saliva and tooth/gum) did not observe changes in the gut microbiota taxonomic composition and diversity over the course of pregnancy, reporting relative stability for all four sites [23]. The authors suggest that the differences in study findings may be due to the fact that many mothers in the Koren et al., study were in receipt of a dietary intervention between T1 and T3. Further studies investigating the gut microbiota composition and functionality before, during and post pregnancy in larger cohorts and from different demographics and geographical locations are required.

It is known that excessive weight gain in pregnancy gives way to decreased glucose tolerance and potentially gestational diabetes mellitus (GDM) [24,25]. GDM is associated with adverse pregnancy outcomes including stillbirth, fetal macrosomia, neonatal metabolic disturbances and related issues [26,27]. Furthermore, offspring of mothers with GDM are at greater risk of obesity and diabetes [28]. Medical nutritional therapy is the first-line approach but up to 50% of women fail to regain metabolic control by this means and must avail of insulin treatment or hypoglycemic drugs [29,30]. Collado et al. [31] investigated the gut microbiota during pregnancy in overweight and normal weight women and reported that *Bacteroides* and *Staphylococcus* were significantly higher in overweight women, and mother’s weight and BMI before pregnancy correlated with higher levels of *Bacteroides*, *Staphylococcus* and *Clostridium*. In both normal weight and overweight women, bacterial counts increased from T1 to T3. In another study, overweight or obese mothers presented gut microbiota with lower alpha diversity compared to lean mothers four days after delivery [32]. Most of the taxa that differentiated the two groups were higher in the lean mothers and included *Parabacteroides*, *Lachnospira*, *Faecalibacterium prausnitzii*, *Christensenellaceae* family members, *Rumincoccus* and *Bifidobacterium*, all of which have shown consistent associations with leanness. These maternal gut microbiota characteristics were not associated with overall differences in the infant gut microbiota over the first two years of life but the authors state that the presence of specific OTUs in the maternal gut microbiota at the time of delivery increased the chances of being present in the infant gut at 4–10 days old which included some lean-associated taxa. Further research is required to determine the degree to which these maternal microbial differences influence the health of the infant over time. More recently, Crusell et al. [33] reported that the gut microbiota of pregnant women with GDM differed substantially from normoglycaemic pregnant women in T3. At phylum level, Actinobacteria was observed to be more abundant in GDM women, while at genus level *Collinsella*, *Rothia* and *Desulfovibrio* were more abundant. The normoglycaemic pregnant women showed enrichment of *Faecalibacterium*, *Anaerotruncus* and depletion of *Clostridium* (sensu stricto) and *Veillonella*. Regardless of metabolic status, OTU richness and Shannon index decreased from late pregnancy to postpartum, reflecting an observation of Koren et al. [21]. *Christensenella* OTUs were associated with higher fasting plasma glucose concentration, while OTUs assigned to *Akkermansia* were associated with lower insulin sensitivity. Eight months after delivery, the microbiota of women with GDM during pregnancy was still aberrant in terms of composition resembling the aberrant microbiota composition of non-pregnant individuals with type 2 diabetes. Further studies are required to determine if such microbiota disruption places these individuals at increased risk of developing type 2 diabetes.

This topic has been further reviewed by Ponzo et al. [29] who also reviewed the potential of the microbiota as a therapeutic target in GDM and concluded that certain microbiota-accessible carbohydrates (MACs) could beneficially modulate the gut microbiota and hence host metabolism in GDM patients. For example, reduced abundance of *Bacteriodes* by the end of pregnancy was reported for women with GDM who consumed higher intakes of oligosaccharides and fibre [34]. This is of significance given that the genus is associated with overweight in pregnancy [31]. In a randomized placebo-controlled clinical trial involving 52 pregnant women in T3, consumption of a synbiotic composed of *Lactobacillus sporogenes* and a prebiotic mixture daily for nine weeks resulted in significantly decreased serum insulin levels and beneficially impacted other insulin actions but did not affect fasting plasma glucose levels and serum high-sensitivity C-reactive protein [35]. More recently, a synbiotic composed of fructooligosaccharide (FOS) and a mixture of probiotic lactobacilli did not influence fasting plasma glucose and insulin resistance/sensitivity indices in women with GDM but proved effective in reducing blood pressure [36].

Interventions involving probiotics alone have generated conflicting results. For example, consumption of the probiotics *Lactobacillus rhamnosus* GG and *Bifidobacterium lactis* BB12 from T1 of pregnancy in a double-blind, placebo-controlled study significantly reduced the incidence of GDM (*P* = 0.003) [37]. However, probiotic supplementation for four weeks (weeks 24 to 28 of gestation) in obese pregnant women did not influence maternal metabolic profile, fasting blood glucose, or pregnancy outcomes [38]. It is possible that the short-term probiotic consumption in this study did not permit the probiotic to induce beneficial changes to the gut microbiota and hence host metabolism. More recently, probiotic supplementation (*L. rhamnosus* GG and *Bifidobacterium animalis* ssp. *lactis*) from T2 of pregnancy to week 28 in overweight and obese women did not prevent GDM [39]. These contradictory results could be due to a number of factors including differences in probiotics and doses used, timing and duration of supplementation as well as differences in host demographics, genetics and baseline gut microbiota of each individual.

Probiotic intervention during pregnancy has been shown to be beneficial for reducing the risk of preeclampsia, a serious condition associated with hypertension and proteinuria that can result in poor pregnancy outcome and is reported to be one of the leading causes of maternal death globally [40,41]. In 2011, a study conducted in Norway reported that regular consumption of milk-based probiotics could be linked with lower risk of preeclampsia in first-time expectant women [40]. A follow-on observational cohort study involving a large sample number of women from both urban and rural regions of Norway of varying ages and socioeconomic status reported that probiotic milk intake in late pregnancy was significantly associated with lower risk of preeclampsia [42]. In the same study, probiotic milk intake during early pregnancy (but not before or during late pregnancy) was significantly associated with lower risk of preterm delivery. However, in the case of both observations, causality could not be established.

Probiotic administration during pregnancy has also generated promising results in terms of treating bacterial vaginosis (as reviewed by Sohn and Underwood, [43]), and infectious mastitis [44,45] and the positive effects of probiotic consumption during pregnancy rendered to the offspring, including prevention of atopic dermatitis [46], eczema and rhinoconjunctivitis [47], have been confirmed in meta-analysis (17 studies, 4,755 children) and a large population-based cohort study (40,614 children), respectively.

Given such promising results, the impact of maternal probiotic supplementation on breast milk composition and the infant microbiome is an important area of research. Breast milk has its own microbiota dominated by members of the staphylococcal and streptococcal genera, but also harbors lactic acid bacteria, bifidobacteria and members of *Propionibacterium* [48]. These microbes originate from the mother’s skin, gut and the infant’s oral mucosa [49]. The transfer of maternal gut microbiota to breast milk is proposed to occur through an entero-mammary route via dendritic cells and macrophages which selectively traffic commensal microorganisms [49,50,51]. Despite this, maternal probiotic supplementation with a fermented milk containing *L. rhamnosus* GG, *Lactobacillus acidophilus* La-5 and *Bif. animalis* ssp. *lactis* Bb-12 four weeks before the expected due date until three months after birth while breastfeeding resulted in the presence of these bacteria in breast milk of only a small subgroup of women and, thus, breastfeeding by probiotic supplemented women is unlikely to be a source of these probiotics in infants [49]. However, a previous study using the same strains reported that probiotic supplementation of pregnant women from 36 weeks of gestation to three months postnatally during breastfeeding reduced the cumulative incidence of atopic dermatitis by almost 40% among offspring at two years of age [52]. Interestingly, a higher prevalence of *L. rhamnosus* GG was found in stool samples of these infants up to three months of age [53]. Simpson et al. [49] suggest that since breastfeeding does not appear responsible for ongoing transfer of *L. rhamnosus* GG to infants, early transfer of *L. rhamnosus* GG may be sufficient to ensure stable colonization in the infant or alternatively children are receiving continued transfer of *L. rhamnosus* GG from their mother via a different route. However, infants from mothers who had consumed the probiotic milk and who did not develop atopic dermatitis during the two years follow-up had reduced T helper (Th) 22 cells at three months of age which may help explain the preventative effects of maternal probiotic supplementation on atopic dermatitis [54]. Consumption of a multistrain probiotic product (VSL#3) by women during late pregnancy and lactation resulted in a significant increase in both lactobacilli and bifidobacteria in colostrum and mature milk in women who underwent vaginal delivery compared with the placebo group, however, analsysis of the bacterial strains and species revealed that the probiotic microorganisms did not pass from the maternal gut to the mammary gland [55]. No significant differences in bifidobacteria and *Lactobacillus* numbers were observed in colostrum and mature milk from mothers who underwent caesarian section from either the probiotic or placebo groups. The authors suggest that a systemic effect may be responsible for the probiotic-dependent modulation of breask milk microbiota in vaginally delivering women.

Interestingly, Kuitunen et al. [56] reported that probiotic supplementation of mothers from week 36 of gestation until delivery altered the immunologic composition of breast milk by significantly increasing IL-10 and significantly decreasing casein IgA antibodies, however, no strong and consistent associations were observed between breast milk antibodies and cytokines and allergy development in children up to the age of five. Baldassarre et al. [57] also reported that high-dose probiotic supplementation during late pregnancy and lactation influenced breast milk cytokine patterns, significantly increasing IL-6 levels in colostrum and IL-10 and TGF-β1 levels in mature breast milk. Furthermore, sIgA levels were higher in newborns whose mothers consumed the high-dose probiotic. A recent study reported that infants born to mothers with depressive symptoms had lower levels of faecal sIgA which could presdispose such infants to higher risk for allergic disease [58]. Thus, probiotic supplementation to mothers during pregnancy could circumvent such low IgA levels in newborns. In contrast, Quin et al. [59] reported that maternal probiotic administration during breastfeeding (from birth to introduction of solid food) did not alter breast milk immune markers. In the same study infants whose mothers were self-administering probiotics also received probiotics directly which resulted in an increase in infant faecal sIgA levels. However, the probiotic group had higher incidences of mucosal-associated illnesses as toddlers. As a consequence the authors caution against probiotic supplementation during infancy until rigorous controlled follow-up studies on their safety and efficacy have been performed although the study itself has a number of limitations including the fact that varying brands and doses of probiotics were consumed by participants.

Studies investigating the impact of prebiotics and synbiotics on breast milk composition and subsequently the infant microbiome are limited. However, Kubota et al. [60] reported that FOS intake (4 g, twice daily) by pregnant and lactating women increased levels of the cytokine IL-27 in breast milk. The consequence of this phenomenon for the onset of allergic disorders in children requires further investigation. A synbiotic consisting of different probiotic strains and FOS administered to lactating mothers for 30 days significantly increased breast milk IgA and TGF-β2 levels and the incidence of diarrhoea in infants whose mother’s were consuming synbiotic was significantly decreased [61]. Synbiotic supplementation to lactating mothers for 30 days was also reported to positively impact mineral levels in breast milk (zinc, copper, iron, magnesium and calcium) which were shown to decrease significantly in the placebo group and the synbiotic also positively impacted infant growth (weight for age Z score and height for age Z score) [62]. Selenium (Se) is an essential trace elemnent for infants and is found in breast milk although its levels can vary depending on the mother’s geographical location due to differences in soil content and hence its accumulation in cereals which are eaten by humans and animals [63]. Taghipour et al. [64] investigated if synbiotic supplementation consisting of FOS and different probiotic strains could increase breast milk Se levels. However, 30 days of synbiotic consumption had no impact on Se levels in breast milk.

Further studies are warranted to fully understand the impact of probiotic/prebiotic/synbiotic supplementation on breast milk composition at the microbiological, immunological and bioactive molecule levels, and to determine the consequence of these changes for both mother and infant in the long term.

### 2.2. Infants

The infant gut microbiota plays an essential role in establishing the gut mucosal barrier, education of the immune system and in preventing enteric pathogen infection [65]. In Part I of this review, we described the development of the infant gut microbiota from birth onwards and while several factors have been shown to influence its composition (host genetics, gestational age, birth mode, feeding regime, antibiotic exposure), the gut microbiota of full-term, vaginally-delivered, exclusively breast-fed infants is generally recognised as representing the healthy microbiota [66,67]. Indeed, owing to its complex mixtures of bioactive components, which change in concentration, structure and function over lactation, human milk is considered the “gold standard” for early life nutrition [68].

In the case of preterm infants, bacterial exposure occurs earlier than normal and antibiotics are frequently administered. Very preterm infants (<32 weeks) and extremely preterm infants (<28 weeks) are at significant risk of sepsis, necrotizing enterocolitis (NEC), feeding intolerance and mortality [69,70]. The preterm infant microbiota has been shown to be lacking in the health-promoting *Bifidobacterium* species and as a consequence of antibiotic administration can be dominated by *Enterobacteriaceae*, *Enterococcus* and *Staphylococcus* [71]. It is also characterised by a lack of microbial diversity [72] and has an increased abundance of Proteobacteria [67]. In a study investigating the distortions in intestinal microbiota development and late onset sepsis in preterm infants, Mai et al. [73] reported that distortions rather than enrichment of potential pathogens were associated with late-onset sepsis. Likewise, no specific pathogen has been identified as responsible for NEC but inappropriate colonisation of the preterm gut has been deemed the causative factor [74]. Preterm infants with NEC have been reported to harbour increased relative abundances of Proteobacteria and decreased relative abundance of Firmicutes and Bacteroidetes prior to the onset of NEC [75,76].

In a recent article, Athalye-Jape and Patole [69] reported that over 25 systematic reviews and meta-analyses of randomized controlled trials involving ~12,000 participants revealed that probiotics significantly reduce the risk of all-cause mortality, NEC ≥ Stage II, late onset sepsis and feeding intolerance in preterm infants and suggest providing probiotics as a standard prophylaxis for preterm infants. In order to gain widespread acceptance, Aceti et al. [77] have pointed out ongoing gaps in the literature and potential directions for future research in relation to probiotic use in preterm infants which include an understanding of the impact of feeding (formula, mother’s milk, donor’s milk) on the relationship between probiotic supplementation and clinical outcome, efficacy of multi-strain probiotics versus single-strain probiotics, safety issues and long-term consequences for such a vulnerable population. However, given the evidence to date it could be argued that it “may be unethical not to treat” with probiotics to reduce the risk of NEC in preterm infants. 

Prebiotics have also proven efficacious for preventing adverse health outcomes in preterm infants. A meta-analysis involving 18 randomized controlled trials consisting of 1322 participants revealed that those in receipt of prebiotics showed significant decreases in incidence of mortality, sepsis, hospital stay duration and time to full enteral feeding; however, there were no differences between control and intervention groups in relation to the morbidity rate of NEC and feeding intolerance [78]. A small number of studies have investigated the efficacy of synbiotics in relation to NEC in preterm infants [79]. In a study involving 400 very low birth weight infants, the rate of NEC was reduced by 2% in the group receiving the probiotic *Bif. lactis*, but was reduced by 4% in the group receiving *Bif. lactis* plus the prebiotic inulin compared to a rate of 12% in the prebiotic group and 18% in the control group [80]. The prebiotic FOS in combination with a probiotic mixture consisting of *L. acidophilus*, *Bifidobacterium longum*, *Bifidobacterium bifidum*, and *Streptococcus thermophilus* significantly reduced the incidence of NEC in preterm infants fed breast milk (2 incidences out of 100) compared to the control group who received breast milk alone (10 incidences out of 100) [81]. In the same study the incidences of Stage II and Stage III (severe) NEC were nil in the test group compared to 5 and 2 cases in the control group, respectively. The incidence of sepsis was also significantly lower in the test group. Likewise, Nandhini et al. [82] reported a 50% reduction in the incidence of NEC of all stages in preterm infants in receipt of a synbiotic consisting of a mix of bifidobacteria and lactobacilli and FOS, however, the severity of NEC, sepsis and mortality were not influenced by synbiotic administration. Despite the apparent success of synbiotics in this small number of studies, a drawback of synbiotics is the difficulty predicting selectivity and specificity and the subsequent mechanisms of action; thus, future studies should focus on unravelling how each component in the mixture, and the mixture as a whole, exerts its (cooperative) effects [79].

Caesarean section has been shown to influence the development and composition of the gut microbiota. In a study involving 192 breast-fed infants, Hill et al. [67] reported that the gut microbiota of the full-term caesarean section infant has a significantly increased faecal abundance of Firmicutes and significantly lower abundance of Actinobacteria compared to the full-term, vaginally delivered infant after the first week of life. A decreased abundance of bifidobacteria has also been reported for six week old infants born by caesarean section [83]. However, the latter study also revealed that this disturbance could be partially restored by exclusive breastfeeding. Likewise, Hill et al. [67] reported that breastfeeding had a beneficial impact on the gut microbiota of infants delivered by caesarean section. With this in mind, it is not surprising that probiotic supplementation to expectant mothers and their infants (for three months) born by caesarean section or receiving antibiotics “benefited” only breast-fed infants in terms of increasing bifidobacteria and reducing Proteobacteria and Clostridia [84].

Probiotic-supplemented infant formula has been on the market in Europe and Asia for over two decades [85]. Such formulae have been shown to result in infant faecal microbiota profiles closer to breast-fed infants [86]. A systematic review of randomized controlled trials up to September 2016 concluded that probiotic-supplemented formulae do not raise safety concerns for healthy infants with regard to growth and adverse effects, however, while some beneficial effects are possible (reduction in number of episodes of gastrointestinal infection, diarrhoea and respiratory symptoms, lower frequency of colic or irritability and better growth) the review concluded there was a lack of robust clinical evidence to recommend their routine use albeit this could be due to the small amount of data on specific probiotic strain(s) and their outcomes rather than an authentic lack of an effect [87]. With this in mind, a meta-analysis conducted in 2018 investigated the efficacy of a single probiotic strain, namely *Lactobacillus reuteri* DSM17398 to treat infant colic [88]. Four double-blind trials with 345 colic infants were included. The study concluded that the probiotic strain in question is effective for treating colic but only in breast-fed infants. With regard to formula-fed infants, the intervention effects were insignificant, however, the authors state that there were insufficient data to make conclusions and thus there is a critical need for more rigorous randomized controlled trials with this strain in formula-fed infants suffering from colic.

The most common prebiotics used in infant formulae include a 9:1 mixture of short chain galactooligosaccharides (GOS) and long chain FOS [89]. A systematic review of 41 randomized controlled clinical trials concluded that feeding prebiotic-supplemented infant formulae to healthy infants is safe in terms of adverse effects and growth [89]. The primary beneficial effect was stool softening but no robust evidence exists to recommend prebiotic-supplemented formulae. As in the case of probiotics, the lack of sufficient data on specific prebiotics was possibly responsible for this conclusion.

A systematic review involving three randomized controlled clinical trials (*n* = 475) on the efficacy of synbiotic-supplemented formulae in 2012 concluded that while synbiotics increased stool frequency they had no impact on stool consistency, colic, spitting up/regurgitation, crying, vomiting or restlessness [90]. However, a recent study showed that amino acid-based formula supplemented with *Bifidobacterium breve* M-16V and FOS over 26 weeks was capable of significantly increasing faecal percentages of bifidobacteria and reducing the *Eubacterium*/*Clostridium coccoides* group in infants with non-IgE-mediated cow’s milk allergy (*n* = 35) [91]. Interestingly, reported ear infections and use of dermatological medication were also significantly lower in the synbiotic group. A synbiotic starter formula containing *Bif. lactis* and FOS fed to 280 infants of age 0.89 months over a three-month period significantly reduced infantile crying and colic, functional constipation and daily regurgitation compared to the reported median prevalence for a similar age according to the literature [92]. Feeding a synbiotic-supplemented formula to infants who had been completely weaned from breast milk to infant formula at 28 days of age until 12 months of age resulted in a significant reduction in the cumulative incidence of lower respiratory tract infections compared to the prebiotic group but as the confidence interval of the estimate was wide, the authors suggest uncertainty with regards to this result [93]. The synbiotic in this case consisted of FOS, GOS and *Lactobacillus paracasei* ssp. *paracasei* F19. Feeding caesarean-born infants formula supplemented with *Bif. breve* M-16V and FOS and GOS from birth until week 16 generated a bifidogenic effect that lasted until week 8, thus emulating the gut physiological environment of vaginally-delivered infants, and reduced *Enterobacteriaceae* until week 12 [94].

These studies suggest that probiotics, prebiotics and synbiotics have a beneficial role to play in infant nutrition, and particularly in vulnerable infants including preterm and those born by caesarean section or for those for who breast milk is not an option. However, in order to incite greater confidence in both the medical profession and the public in general there is a need for large cohort, possibly multi-centre randomized controlled trials that focus on specific prebiotics, probiotics and synbiotics which assess their impacts and modes of action on the gut microbiota, infant health and wellness and the long-term outcomes for these parameters.

### 2.3. Elderly in Nursing Homes

In Part I of this review we discussed the elderly (> 65 years) microbiota which is generally characterised by a reduction in microbial diversity, a decrease in species associated with short chain fatty acid (SCFA) production, especially butyrate, an increase in opportunistic pathogens [95,96] and even greater inter-individual variation than observed in adults [97]. The gut microbiota of those in long-stay residential care facilities is significantly less diverse than individuals of the same age group who reside within the community and the increased frailty observed in long-stay care residents correlates with loss of community-associated microbiota [98]. In the same study, the distinct microbiota groups identified as a result of residence location also overlapped with diet where individuals in long stay care facilities tended to consume high fat, low fibre diets versus the low fat, high fibre diet of community dwellers. Furthermore, scientists are hypothesizing that the gut microbiota may influence sarcopenia through a gut-muscle axis, a syndrome which affects older individuals (recently reviewed by Ticinesi et al. [99,100]). Sarcopenia is described as depletion of muscle mass and reduction of muscle performance which both result from anabolic resistance or boosted protein catabolism [101]. It is distinct from frailty although the two may overlap [102]. To date, there have been no studies in humans investigating the microbiome of sarcopenic individuals; however, Siddharth et al. [103] identified a distinct faecal microbiota composition associated with age-related muscle wasting in rats which revealed a reduction in several taxa reported to have pro-anabolic and anti-inflammatory properties. Interestingly, the SCFA butyrate was shown to have beneficial effects on muscle mass in ageing mice, partially or wholly protecting them from muscle atrophy [104], and the human commensal *L. reuteri* inhibited muscle wasting in mice [105].

Osteosarcopenic obesity describes an impairment in muscle, bone and adipose tissue which occurs in elderly individuals in conjunction with an altered gut microbiota, especially in those in long-term care facilities [106]. The increased adiposity associated with osteosarcopenic obesity can manifest as overt clinical overweight/obesity, redistribution of fat around visceral organs or the infiltration of fat into muscle and bone tissues, thus impairing their function [106]. It is more prevalent in older women than older men and women with osteosarcopenic obesity have decreased strength, balance and mobility compared to those with obesity, osteoporotic obesity and sarcopenic obesity alone [107]. The gut microbiota has been shown to regulate bone mass in mice [108] and the probiotic *L. reuteri* was reported to protect menopausal ovariectomized mice from bone loss [109].

In Part I of this review, we discussed the obese gut microbiota and the link between the gut microbiota and energy storage in the body [2]. Given the reported links between the gut microbiota, muscle, bone, and adipose tissues, such studies suggest that the gut microbiota could be a therapeutic target in the treatment of sarcopenia and osteosarcopenic obesity and aid in the prevention of associated outcomes such as increased risk for falls, fractures, long-term frailty and immobility [106]. This is an exciting area in microbiome research and may have profound implications for the ageing process. The gut-muscle axis is further discussed in Section 2.4 (Physical Activity).

While the nutritional needs of the elderly do not vary significantly from younger adults with similar caloric expenditure and anthropometric and physiological features, elderly individuals are at greater risk of malnutrition [110,111] owing to a number of factors outlined in Part I of this review [2]. Indeed, it has been reported that approximately 30% of individuals over 50 years of age do not consume the RDA for protein [110,112]. Other nutrients which fall short in this demographic include fibre, iron, vitamins D, B_6_ and B_12_ and folic acid [110,113].

Salazar et al. [110] suggest that nutritional strategies for the elderly should not just focus on nutritional deficiencies but also consider the intestinal microbiota and immune function. With this is mind, the following have been considered relevant targets for interventions in this age group: (1) reduced microbial diversity, (2) low-levels of butyrate-producing bacteria, (3) imbalanced proportions and reduced levels of SCFA, (4) increased incidence of *Clostridium difficile* infection, (5) higher levels of lactate, (6) methane, and (7) branched chain fatty acids (valeric, isovaleric, isobutyric and caproic acids) [95,97,110,111,114,115,116,117,118]. For the purpose of this review, we have focused on the impact of interventions in this age group involving fibre, prebiotics, probiotics and synbiotics.

Bahgurst et al. [119] investigated the long-term (12-month) effects of moderate fibre supplementation (an increase in fibre intake of ~70%) in a nursing home population, of mean age 83 years, with an emphasis on bowel function, body weight and mineral status. As well as improving bowel function, the fibre supplementation improved nutrient density of the diet without increasing body weight. In a more recent study, potato intake in 32 institutionalised elderly subjects (aged between 76 and 95 years) was directly associated with faecal SCFA concentrations, and apple intake was directly associated with propionate concentration [120]. In the same study, cellulose intake was associated with acetate and butyrate concentrations. While the sample size was low, the approach provides an opportunity to generate improved diets with an emphasis on increasing specific or total SCFAs.

Probiotic consumption in the elderly cohort has been shown to improve certain immune parameters as well as beneficially modulating the intestinal microbiota. The immuno-stimulating probiotic *Bif. lactis* HN019 enhanced immunity in elderly subjects aged 68 to 84 years following consumption of either 5 × 10^10^ microorganisms/day or 5 × 10^9^ microorganisms/day for three weeks [121]. Daily consumption of a probiotic mixture composed of *Lactobacillus gasseri* KS-13, *Bif. bifidum* G9-1 and *Bif. longum* MM-2 for three weeks by elderly participants (70 ± 1 year) increased IL-10 concentrations compared to the placebo [122]. In addition, 48% of participants in the probiotic group had increased faecal bifidobacteria compared to 30% in the placebo which was significantly different (*P* < 0.05). Moreover, 55% of participants in the probiotic group had increased lactic acid bacteria and 52% had decreased *E. coli* compared to 43% and 27% in the placebo group, respectively, representing significant differences (*P* < 0.05). Bacterial groups matching the butyrate producer *F. prausnitzii* were also more abundant in stool samples from the probiotic group. The overall changes resembled those observed in healthy younger populations. Gao et al. [123] reported a similar finding in relation to *F. prausnitzii* levels following long-term probiotic consumption by an elderly cohort. While consumption of a probiotic cheese containing *L. rhamnosus* HN001 and *L. acidophilus* NCFM by an elderly population increased the numbers of said probiotics in faeces, there was no effect on faecal immune markers [124]. However, the probiotic cheese was associated with a trend towards lower *C. difficile* counts, an effect which was statistically significant in the subpopulation that were found to harbor *C. difficile* at the beginning of the study. Likewise, consumption of one probiotic-containing biscuit (*Bif. longum* Bar33 and *Lactobacillus helveticus* Bar13) per day for one month was found to revert the age-related increase in the following opportunistic pathogens, *C. difficile*, *Clostridium* cluster XI, *Clostridium perfringens*, *Enterococcus faecium*, and the enteropathogenic genus *Campylobacter* in elderly volunteers [125]. Consumption of a fermented oat drink containing *Bif. longum* 46 and *Bif. longum* 2C by elderly nursing home residents for six months significantly increased faecal bifidobacteria levels [126]. In an attempt to understand how probiotic consumption in the elderly promotes health, Eloe-Fadrosh et al. [127] reported the impact of a single probiotic strain (*L. rhamnosus* GG) on the structure and functional dynamics of the gut microbiota in healthy elderly individuals following consumption of 10^10^ colony forming units (cfu) twice daily for 28 days. The probiotic modulated the gut microbiota transcriptome. In particular, *Bifidobacterium* genes involved in flagellar motility, chemotaxis and adhesion were increased following probiotic consumption, and gene expression in the butyrate producers *Ruminococcus* and *Eubacterium* was also increased. This suggests that this single probiotic strain has the potential to promote anti-inflammatory pathways.

Prebiotic supplementation in the elderly has generated promising results in terms of beneficial alterations to the gut microbiota and also frailty syndrome. Daily consumption of 8 g of short chain FOS for four weeks by healthy elderly individuals led to increases in faecal bifidobacteria counts [128]. Daily doses of GOS at 5.5 g for four weeks in an elderly group resulted in significant increases in bifidobacteria and bacteroides and immune alterations which included lower IL-1β levels and higher C-reactive protein, IL-10, IL-8 and natural killer cell activity [129]. Most recently, prebiotic supplementation which involved a mix of prebiotics at 20 g/day for 26 weeks to frail elderly subjects did not induce global changes in gut microbiota alpha and beta diversity but the abundance of certain bacterial taxa increased including *Ruminococcaceae* and the levels of the chemokine CXCL11 were significantly reduced [130]. This particular chemokine is produced in response to microbial antigens [131]; although the authors state that the health/clinical benefits are not clear. Buiges et al. [132] investigated the impact of prebiotic supplementation on frailty syndrome in elderly individuals in a randomized, double-blind clinical trial. In this case, the prebiotic in question, Darmocare Pre^®^ which is a mix of inulin and FOS did not significantly modify the overall rate of frailty but did significantly improve two frailty criteria, exhaustion and handgrip, following 13 weeks of daily consumption. The authors suggest that therapeutics aimed at the gut microbiota–muscle–brain axis should be considered for the treatment of frailty syndrome. More recently, the same prebiotic was tested in nursing home residents and of the 28 participants in the intervention group, 25 revealed reduced frailty index levels where the moderately/severe frail participants showed the greatest reduction [133].

In a prospective, double-blind, placebo-controlled, randomized single centre study involving 40 healthy elderly subjects (aged between 60–80 years), intake of a synbiotic combination of soluble corn fibre with *L. rhamnosus* GG for three weeks tended to promote innate immunity in elderly women and 70- to 80-year-old volunteers (male and female) by increasing natural killer cell activity [134]. Interestingly, the pilus-deficient version of *L. rhamnosus* GG, termed *L. rhamnosus* GG-PB12, with the soluble corn fibre increased natural killer cell activity in older volunteers compared to soluble corn fibre alone. The combination of *L. rhamnosus* GG-PB12 with the corn fibre also decreased C-reactive protein, an indicator of inflammation in the body. Total cholesterol and LDL-cholesterol was also reduced in individuals who had presented with elevated levels following intake of *L. rhamnosus* GG with the soluble corn fibre. The genus *Parabacteroides* was significantly increased as a result of either strain with the corn fibre. Soluble corn fibre alone and soluble corn fibre with *L. rhamnosus* GG increased levels of *Ruminococcaceae incertae sedis*. Decreases in the levels of *Ruminococcaceae* and *Parabacteroides* have been pinpointed as the main microbial shifts associated with ageing in mice [135,136]. Slight reductions were observed in *Oscillospira* (positively associated with leanness and health [137]) and the sulphate-reducing *Desulfovibrio* following *L. rhamnosus* GG with soluble corn fibre consumption, whereas only *Desulfovibrio* decreased following intake of *L. rhamnosus* GG PB12 with corn fibre.

These studies indicate that dietary interventions involving fibre, prebiotics and probiotics in the elderly, and especially those in residential care, can induce beneficial changes to the gut microbiota with potential to improve immune function and gut homeostasis. The gut microbiota of healthy younger adults is considered a suitable reference for the elderly microbiota assuming the younger population shares the same geographical location, historical past and social habits/lifestyle etc. [111,138]. Thus, further studies are required for this cohort to find interventions which can improve the relevant targets of the intestinal microbiota and immune function and generate meaningful physiological changes which translate to improved general health and well-being (e.g., frailty reduction, improved mobility, reduced risk of fracture and falls, improved sleep and overall energy levels etc).

### 2.4. Physical Activity

The impact of exercise on the gut microbiota has only begun to be studied in recent years. In a first study of its kind, Clarke et al. [139] reported increased microbial diversity in a professional rugby team of a preseason camp compared to age-matched and BMI-matched controls. The gut microbiota differences observed in these athletes correlated with protein consumption and creatine kinase, a marker of extreme exercise. In fact, protein accounted for 22% of the total energy intake of athletes compared to 16% in the low BMI control group and 15% in the high BMI control group. A follow-on study investigating the metabolic activity of the gut microbiota of these athletes revealed several differences compared to the control groups [140]. Pathways involved in amino acid biosynthesis, carbohydrate metabolism and antibiotic biosynthesis were increased in athletes. SCFA levels were also increased in the athletic group. Of note, athletes also excreted higher levels of the uremic toxin, trimethylamine-*N*-oxide (TMAO), which has been discussed in Part 1 of this review [2] as it has been proposed as a risk factor for cardiovascular disease in humans. However, the authors state that the implications of this result are limited and require further study. As expected, the athletes consumed more calories and macronutrients than the control groups. Fibre intake was also higher in the athletic group compared to the high BMI control group.

In order to better understand the impact of exercise on the gut microbiota, Estaki et al. [141] analysed the microbiota of healthy individuals with varying levels of fitness and reported that cardiorespiratory fitness correlated with increased microbial diversity in healthy humans. Six weeks of endurance exercise by overweight women was reported to alter the gut metagenome with an increase in the health-promoting *Akkermansia* and a decrease in Proteobacteria [142]. Notably, diets did not change during the six weeks control period before the exercise intervention or during the six-week exercise period. Despite the changes to the gut microbiota, systemic metabolites and body composition were not greatly affected. Likewise, five-weeks of endurance exercise by elderly men was found to significantly decrease the relative abundance of *C. difficile* and significantly increase *Oscillospira* which correlated with beneficial changes in several cardiometabolic risk factors [143]. Changes in food intake did not differ between control and exercise periods. Allen et al. [144] reported that six weeks of endurance exercise increased faecal SCFA concentrations in lean but not obese participants and the metabolic changes were associated with changes in bacterial taxa and genes capable of producing SCFAs. Interestingly, the exercise-induced changes were reversed when exercise ceased. While these studies demonstrate the beneficial impacts of exercise on the gut microbiota, exercising to the point of exhaustion may induce detrimental changes [99]. For example, intense military training undertaken by soldiers for four days resulted in increased intestinal permeability and changes in microbiota composition which included increased alpha diversity and an increase in the abundance levels of potentially pathogenic taxa (e.g., *Staphylococcus*, *Peptostreptococcus*, *Peptoniphilus*, *Acidaminococcus*, and *Fusobacterium*) at the expense of several taxa thought to protect against pathogen invasion (e.g., *Bacteroides*, *Faecalibacterium*, *Collinsella*, and *Roseburia*) [145]. Thus, as suggested by Ticinesi et al. [99], the impact of exercise on the gut microbiota may depend on the intensity and duration, however other confounders should also be considered including diet, nutrient intake and body composition parameters, a topic that requires further investigation.

We have already mentioned the gut–muscle axis hypothesis in Section 2.3 (Elderly in Nursing Homes) and indeed it has been proposed that the gut-microbiota axis may be two-way with exercise influencing the microbiota and the microbiota influencing muscle [99], the latter of which was observed in the case of muscle-wasting in rats [103]. Ticinesi et al. [99] provided a list of hypothesised pathways linking gut microbiota modulation to muscle function and include (1) bioavailability of dietary proteins and specific amino acids, (2) vitamin synthesis such as folate, B_12_ and riboflavin, (3) biotransformation of nutrients such as polyphenols and ellagitannins, (4) intestinal mucosa permeability, (5) bile acid biotransformation, (6) SCFA synthesis. In the case of intestinal dysbiosis, changes in these pathways may have negative consequences for skeletal muscle function. The interaction between the gut microbiota and the immune system is also another factor in the gut–muscle axis hypothesis [99] given the purported links between inflammation and age-related muscle wastage [146]. Further studies in this field are clearly warranted to understand the complex relationships between all these factors. Ultimately, this should help in the design of strategic exercise programmes, diets and probiotic/prebiotic interventions which are optimised for life stage ensuring a healthy gut microbiota for optimal skeletal-muscle function and host health.

Nowadays probiotic supplementation is common practice for many athletes involved in different sports and is generally taken to reduce incidence of infection, especially upper respiratory tract infections and gastrointestinal problems. Upper respiratory illness is reported to account for 35%–65% of illness presentations to sports medicine clinics [147]. These infections are generally caused by common respiratory viruses, allergic responses to aeroallergens and exercise-related trauma to respiratory epithelial membrane integrity [147]. Gastrointestinal disorders in athletes can occur during or after intense physical activity and include bloating, abdominal pain, diarrhoea, and blood in the stool and may be caused by inadequate blood supply to the digestive tract during exercise [148,149]. Gastroesophageal reflux disease (GERD) can also be exacerbated by intense exercise [150].

Interestingly, probiotic supplementation in the form of *Lactobacillus casei* Shirota to men and women (*n* = 32) involved in endurance-based physical activities for four months of the winter significantly reduced the incidence of upper respiratory tract infections compared to the placebo group and the proportion of placebo subjects who experienced one or more weeks with upper respiratory tract infection symptoms was 36% higher than those taking the probiotic [151]. Salivary IgA was also significantly higher in the probiotic group, an effect which was not evident at baseline. In a later study with the same probiotic strain, five months of supplementation to university athletes and game players (*n* = 243) had no impact on upper respiratory tract infection symptoms which the authors state could be attributable to the low incidence of such symptoms during the study [152]. The probiotic was associated with plasma cytomegalovirus and Epstein Barr virus antibody titres which could be interpreted as an improvement in immune status. Consumption of heat-killed *Lactococcus lactis* JCM 805, also known as LC-Plasma, for 13 days was shown to relieve the morbidity and symptoms of upper respiratory tract infections in male athletes performing high-intensity exercise [153]. This was achieved by activation of plasmacytoid dendritic cells (pDC) which are known to play a significant role in viral infection. Furthermore, the bacterial strain decreased fatigue accumulation during consecutive high intensity exercise. A later study in mice showed that LC-Plasma-activation of pDC in turn attenuates the concentration of fatigue controlled cytokine TGF-β and muscle degenerative genes [154]. Consumption of a probiotic powder containing *L. rhamnosus* GG and *Bif. animalis* ssp. *lactis* BB12 reduced the duration and severity of upper respiratory tract infections in college students and fewer school days were missed [155].

Administration of *Lactobacillus fermentum* (PCC^®^) to male (*n* = 64) and female (*n* = 35) competitive cyclists for 11 weeks generated mixed results in terms of gastrointestinal illness and lower respiratory illness symptoms [156]. Males experienced a reduction in the severity of gastrointestinal illness which became more pronounced as training load increased. The load of lower respiratory illness symptoms was also reduced in males compared with the placebo but actually increased in females on the probiotic. Probiotic numbers increased 7.7-fold more in males compared to an unclear 2.2-fold increase in females. Thus it was concluded that *L. fermentum* could be a useful nutritional adjunct for exercising males. Consumption of *Lactobacillus salivarius* for four months in the spring by both men and women (*n* = 66 in total) participating in endurance-based physical activities had no impact on incidence of upper respiratory tract infections or mucosal immune markers [157]. While probiotic supplementation for one month did not have any effect on severity of upper respiratory tract infections or gastrointestinal episodes in 30 elite rugby union players, it did significantly reduce the number of participants experiencing such symptoms and tended to reduce the number of illness days compared to placebo [158]. Consumption of a multispecies probiotic for three months in the winter by trained athletes (*n* = 33) reduced the incidence of upper respiratory tract infections compared to the placebo and reduced exercise-induced tryptophan-degradation rates [159]. Despite this, probiotic supplementation did not improve athlete performance. The probiotic *L. helveticus* Lafti L10 significantly reduced the duration of upper respiratory tract infection episodes in 39 elite athletes during 14 weeks of supplementation in the winter but did not influence severity of symptoms or incidence [160]. A follow-on study indicated that the probiotic modulated mucosal and humoral immunity in elite athletes [161]. The probiotic was also shown to exert certain antioxidant potential in elite athletes following three months of supplementation but further research is warranted to confirm this effect [162]. Interestingly, based on the results of a meta-analysis of randomized controlled trials comparing probiotics with placebo to prevent acute upper respiratory tract infections in children, adults and older people (*n* = 3720), Hao et al. [163] concluded that probiotics were better than placebo for reducing the incidence of such episodes, the duration of episodes as well as cold-related school absence and antibiotic use. A recent systematic review of the effects of probiotic supplementation on physically active individuals (*n* = 1680, athletes and non-athletes) concluded that positive effects were reported for several outcomes including respiratory tract infection, markers of immunity and gastrointestinal symptoms; however, the study failed to identify standardised supplementation protocols owing to the distinct protocols employed across the studies, as well as different measured outcomes and small sample size [164].

In terms of performance, supplementation with certain probiotics has been shown to have a beneficial effect by presumably influencing host and nutrient metabolism. For example, taking *Lactobacillus plantarum* TWK10 for six weeks resulted in significantly higher endurance performance and glucose content in a maximal running treadmill test in eight adults compared to the placebo group (*n* = 8) such that the authors suggest it could have potential as an aerobic exercise supplement [165]. *L. plantarum* PS128 was reported to have beneficial effects on high-intensity, exercise-induced oxidative stress, inflammation and performance in a study involving triathletes [166].

Very few studies have investigated the impact of prebiotics on athletes. A multi-strain probiotic/prebiotic antioxidant intervention for 12 weeks in recreational athletes prior to a long-distance triathlon was shown to reduce plasma endotoxin unit levels and maintain intestinal permeability [167]. Gastrointestinal symptoms such as cramping, diarrhoea, nausea and abdominal pain etc. were also significantly lower in the test group compared with the placebo during the intervention.

In efforts to generate probiotic and prebiotic supplements for physically active individuals, the type and intensity of exercise performed should be taken into account considering that exercise exerts its own effects on the gut microbiota and may even influence the efficacy of the intervention, although this has yet to be investigated. Furthermore, marketing of such supplements should make clear the intended beneficial effects which range from improved immunity against particular illnesses to improved performance. In this regard, double-blind, randomized controlled, multi-centre trials involving larger cohorts of participants with standardised supplementation protocols are required.

In terms of diet, athletes tend to consume more protein than the average population and an early research review in 1984 examining the importance of protein for athletes concluded that athletic individuals should consume 1.8 to 2.0 g of protein/kg of body weight/day which is approximately twice that recommended for sedentary individuals [168]. However, the studies reviewed in Part I of this review [2] in relation to the impact of protein on the gut microbiota clearly showed that dietary source is a critical factor with animal- and plant-derived protein sources generating heterogeneous responses in terms of gut microbiota composition and functionality. Further research is warranted to fully comprehend the consequences of these alterations but Blachier et al. [169] concluded that some caution should be exercised around high protein diets given their effects on the gut microbiota following a review of the topic. In this regard, probiotic supplements geared at the sports industry should be investigated with respect to typical dietary extremes undertaken by athletic individuals.

### 2.5. Stress

In a recent report by a UK charity called the Mental Health Foundation, stress is defined as the “body’s response to pressures from a situation or life event” [170]. According to this report almost 74% of people surveyed from a total of 4169 adults felt stressed to the point of being overwhelmed or unable to cope at some point of 2018. Stress can be caused by a variety of events and life situations. Some common stress triggers include workplace related stress, exam stress, and illness. Exam stress can be a major issue for students, negatively affecting sleep patterns and academic performance [171,172]. With regard to work-related stress, a report compiled by the UK Health and Safety Executive stated that over half a million people suffer from work-induced stress, depression or anxiety, resulting in a loss of 15.4 million working days over 2017 and 2018 [173]. Workplace stress is also the major source of stress for adults in the USA [174]. Implementation of work-place policies and procedures is critical in tackling these issues. However, we now know that quality of diet, specific dietary components and supplements can aid in the treatment or prevention of depression, anxiety and stress symptoms [175]. Opie et al. [176] compiled a number of dietary recommendations for the prevention of depression based on current available evidence which included increased consumption of fruits, vegetables, wholegrain cereals, legumes, nuts and seeds, high consumption of omega-3 polyunsaturated fatty acids, limited intake of processed foods and replacement of unhealthy foods with nutritious wholesome foods. In addition, the study recommended following traditional dietary patterns such as the Japanese, Norweigan or Mediterranean diet, the latter of which has been discussed in detail in Part I of this review in terms of its beneficial impacts on the gut microbiota and host health [2]. Thus, the impact of diet on the gut microbiota undoubtedly influences our emotional state. Even in adults without diagnosed mood disorder, gut microbes have been shown to be connected to mood (depression, anxiety and stress) and these relationships differ by sex and are influenced by dietary fibre intake [177]. It is now known that the gut microbiota communicates with the brain along the brain–gut–microbiota axis as evidenced from preclinical and some clinical studies [178]. In Part I of this review [2], we discussed the ability of the gut microbiota to produce neurochemicals including gamma amino butyric acid (GABA), a major inhibitory neurotransmitter in the brain [179], as well as its involvement in serotonin biosynthesis [180] and tryptophan metabolism [181].

The impact of psychological stress on the gut microbiota has been reviewed recently by Karl et al. [182]. To date, most studies have focused on rodent models, many of which have demonstrated a reduction in *Lactobacillus* following exposure to stress [183,184,185,186,187]. Interestingly, exam stress in humans has been shown to reduce gut lactic acid bacteria [188] and Taylor et al. [177] reported an inverse relationship between Anxiety scale scores and *Bifidobacterium* in females, while in males an inverse relationship was observed between depression-scale scores and *Lactobacillus*. Thus probiotic and prebiotic interventions have the potential to impact the gut-brain axis with beneficial consequences for mood and stress behaviours.

Chronic fatigue syndrome is characterised by persistent and relapsing tiredness and 97% of patients report neurological disturbances resulting in a variety of emotional symptoms of which anxiety and depression are the most common [189,190]. In a pilot study involving 39 chronic fatigue syndrome patients intake of *L. casei* strain Shirota for two months resulted in a significant decrease in anxiety symptoms compared with the control group (*P* = 0.01) [189]. *Lactobacillus* and bifidobacteria counts were also significantly increased as a result of probiotic administration. A probiotic mix consisting of *L. helveticus* R0052 and *Bif. longum* R0175 was found to relieve psychological distress significantly in healthy human volunteers (*n* = 55) participating in the clinical trial as measured by the Hopkins Symptom Checklist, the Hospital Anxiety and Depression Scale, the Coping Checklist and urinary free cortisol [191]. Black Depression Inventory scores were reduced in volunteers (*n* = 20) with major depressive disorder following eight weeks of supplementation with a probiotic mixture consisting of *L. acidophilus*, *L. casei* and *Bif. bifidum* [192]. Several metabolic parameters were also improved including serum insulin levels and homeostasis model assessment of insulin resistance. Interestingly, probiotic administration has also proven beneficial in the case of postpartum symptoms of depression. In this case, 423 women participated in the trial at 14–16 weeks of gestation and consumed *L. rhamnosus* HN001 daily until six months postpartum [193]. Mothers in the probiotic group reported significantly lower depression scores and anxiety scores compared to mothers in the placebo group in the postpartum period.

In terms of exam stress, consumption of fermented milk containing *L. casei* Shirota for eight weeks by healthy medical students (*n* = 24) until the day before examination resulted in significantly reduced salivary cortisol levels and plasma tryptophan levels compared with the placebo group (*n* = 23) and two weeks after the examination the probiotic group had significantly higher faecal serotonin levels [194]. Furthermore, during the pre-examination period at 5–6 weeks, the rate of subjects experiencing common abdominal and cold symptoms and total number of days experiencing such symptoms was significantly lower in the probiotic group. In rats exposed to water avoidance stress (WAS), the same strain significantly suppressed WAS-induced increases in plasma corticosterone and significantly reduced the number of corticotropin releasing factor–expressing cells in the paraventricular nucleus [195]. In the same study, intragastric administration of the strain, in a dose-dependent manner, stimulated gastric vagal afferent activity.

Modulation of the gut microbiota with prebiotics has also generated promising results in terms of emotional symptoms. For example, consumption of the prebiotic trans-GOS for 12 weeks at 7 g/day (but not 3.5 g/day) significantly improved anxiety scores in individuals suffering from irritable bowel syndrome (IBS) compared with the placebo group [196]. Faecal bifidobacteria were significantly increased in the prebiotic group at 3.5 g/day (*P* < 0.005) and 7 g/day (*P* < 0.001). Intake of Bimuno^®^-GOS for three weeks significantly reduced salivary cortisol awakening response in healthy volunteers [197]. In the same study, this particular prebiotic resulted in decreased attentional vigilance to negative versus positive information in a dot-probe task. Consumption of short-chain FOS at 5 g/day for 4 weeks significantly increased faecal bifidobacteria in IBS patients and significantly reduced anxiety scores [198].

While these studies highlight the benefits of particular probiotics, prebiotics and their combinations (summarized in Table 1), the beneficial effects rarely impacted every subject in a test group, albeit they impacted enough to generate statistical significance in most cases. One possible reason is the quality and quantity of an individual’s baseline microbiota. In the following sections, this becomes very apparent whereby studies have begun to disentangle the disparities between responders and non-responders in terms of gut microbiota composition and behaviour, particularly in response to fibre. Moving forward, it is possible that future interventions will have to be individually-tailored following a comprehensive analysis of an individual’s gut microbiome through microbiome testing, the feasibility of which is discussed in Section 6.

## 3. Modifying the Microbiota as A Target for Preventing Over/Undernutrition—Potential of Probiotics, Prebiotics and Dietary Fibre

Undernutrition and overnutrition represent forms of malnutrition which manifest due to imbalances in energy and/or nutrient intake [199]. Symptoms of undernutrition include wasting (low weight-for-height), stunting (low height-for-age) and underweight (low weight-for-age) [199]. Overnutrition results from overfeeding, defined as the supply of energy containing nutrients in excess of requirements resulting in fat storage and other undesirable outcomes as discussed in Part I of this review [2]. Overweight and obesity can coexist with undernutrition, a phenomenon described as the “double burden” of malnutrition by the WHO. Thirteen percent of the world’s population aged 18 years and over are obese [200]. According to the WHO, 462 million adults are underweight and around 45% of deaths among children under five years of age are linked to undernutrition [199]. Given the link between the gut microbiota and energy regulation in the body, probiotics, prebiotics or fibre may provide effective dietary strategies to restore energy homeostasis through strategic manipulation of the gut microbiota.

### 3.1. Probiotics

Several clinical trials have investigated the impact of probiotics on overnutrition in humans. The studies are discussed in this section and are summarized in Table 2.

The bacterium *Bif. breve* B-3 was used in a randomized, double-blind, placebo-controlled trial involving adult volunteers with BMI ranging from 24 to 30 kg/m^2^ [201]. According to the WHO, BMI values from 25.0 to 29.9 represent a pre-obesity nutritional status, while a BMI of 30 falls into class I obesity [205]. In the trial, participants received either placebo (*n* = 25) or a B-3 capsule (*n* = 19) (approximately 5 × 10^10^ cfu/day) for 12 weeks [201]. Consumption of the B-3 capsule significantly lowered fat mass by week 12. Improvements in some blood parameters related to liver function and inflammation were observed and significant correlations could be made between these and the changed fat mass indicating that *Bif. breve* B-3 has the potential to improve metabolic disorders. Since some of the participants in this trial were receiving medication for diabetes, hypertension or hyperlipidemia, another randomized, double-blind, placebo-controlled trial was recently performed with B-3 involving 80 pre-obese adults (25 ≤ BMI < 30) without any disorders [202]. While fat area significantly increased in the placebo group at weeks 4 and 8, no changes were observed in the B-3 group. Indeed, body fat mass and percent body fat were significantly lower in the B-3 group at weeks 8 and 12. The probiotic strain slightly decreased triglyceride levels and improved HDL cholesterol from baseline suggesting potential for the strain to reduce body fat in healthy, pre-obese individuals. In overweight and obese adults, six months consumption of *Bif. animalis* ssp. *lactis* 420 (10^10^ cfu/day) was shown to control body fat mass and reduce weight circumference and food intake [203]. Interestingly, circulating zonulin, a potential marker of intestinal permeability, remained consistently lower in the probiotic group, and changes in zonulin significantly correlated with changes in body fat mass. In addition, changes in high-sensitivity C-reactive protein resembled those of zonulin. Thus, the authors speculate that the probiotic strain exerted its control on body fat mass via circulating zonulin levels and hence gut permeability and by attenuating low-grade inflammation.

Certain probiotic strains have been shown to enhance weight gain to such an extent that they have gained popularity as alternatives to antibiotic growth promoters in animal feed where they are often referred to as direct fed microbials (DFMs) [206]. The mechanisms responsible for this effect include promotion of a favourable gut microbiota, enhanced digestion and absorption of nutrients, altered gene expression in pathogenic microorganisms, and the various mechanistic actions associated with colonisation resistance including immunomodulation [206]. A comparative meta-analysis on the effects of *Lactobacillus* species on weight gain in humans and animals involving 17 randomized clinical trials in humans, 51 studies on farm animals and 14 experimental models concluded that different *Lactobacillus* species exert different effects on weight change and these effects are host-specific, however, *L. acidophilus* administration results in significant weight gain in humans and animals [207]. A more recent systematic review assessing the potential of probiotic diets to significantly influence weight change in obese and non-obese individuals revealed that the effects were species and strain-specific [204]. For example, while *L. gasseri* BNR17 reduced weight gain, *L. gasseri* L66-5 promoted it. A systematic review on the effects of probiotics on child growth involving 12 studies, 10 of which were randomized controlled trials, revealed that probiotics have the potential to improve child growth in children in developing countries and in under-nourished children [208].

Kwashiorkar is a form of severe acute malnutrition (SAM) resulting from inadequate nutrient intake coupled with additional environmental insults [209]. By studying Malawian twin pairs during the first three years of life, of which half of the twin pairs remained well-nourished, 43% became discordant and 7% manifested concordance for acute malnutrition, Smith et al. [209] revealed the gut microbiota as a causative factor since the kwashiorkor microbiome with Malawian diet induced marked weight loss when transplanted to mice. Million et al. [210] reported a dramatic depletion of obligate anaerobes in SAM. Indeed, while *Enterococcus faecalis*, *E. coli* and *Staphylococcus aureus* were consistently enriched in cases of SAM, several species of the following families were consistently depleted: *Bacteroidaceae*, *Eubacteriaceae*, *Lachnospiraceae*, and *Ruminococcaceae*, along with dramatic depletion of *Methanobrevibacter smithii*. Overall, total bacterial number was decreased and faecal redox potential increased. Such microbes have been termed the healthy mature anaerobic gut microbiota (HMAGM) [211]. Indeed, the first step in gut microbiota alterations associated with SAM is early depletion of the pathogen inhibitor *Bif. longum*, followed later on by absence of the HMAGM resulting in deficient energy harvest, immune protection and vitamin biosynthesis which are associated with malabsorption, systemic pathogen invasion and diarrhoea [211]. In this regard, Alou et al. [212] used a combination of culturomics and metagenomics to analyse the stool samples of healthy children and kwashiorkor patients to identify potential probiotics to treat SAM. This resulted in the identification of 12 species in healthy children which were absent in kwashiorkor patients. These 12 potential probiotics represent an array of possible functions including antibacterial potential, polysaccharide fermentation, butyrate production, antioxidant potential or simply common members of the gut microbiota from healthy humans and healthy breast-fed infants. The authors propose that this cocktail of probiotics offers a defined, reproducible, safe and convenient alternative to faecal transplantations for the treatment of SAM in children.

Furthermore, it is important to mention that probiotic-mediated beneficial effects may not require live cells. Indeed, Plovier et al. [10] reported that a purified membrane protein from *Akkermansia muciniphila*, or the pasteurised bacterium, improved metabolism in diabetic and obese mice. Indeed, pasteurisation enhanced its capacity to reduce dyslipidemia, fat mass development and insulin resistance. This finding suggests that the beneficial effects of difficult-to-cultivate microorganisms may still be harnessed for therapeutic use by using dead or injured cells.

### 3.2. Prebiotics

Parnell and Reimer [213] investigated the impact of daily oligofructose supplementation (21 g/day) in healthy adults with BMI > 25 for 12 weeks in a double-blind, placebo-controlled trial. Compared to the control group, which experienced an increase in weight gain (0.45 ± 0.31 kg), the prebiotic group experienced a 1.03 ± 0.43 kg loss in body weight. Glucose regulation was also improved in the prebiotic group who self-reported a reduction in caloric intake. The authors suggest that the suppression in ghrelin expression and enhanced peptide YY (PYY) expression observed in the prebiotic group partly contributes to the reduction in energy intake. In overweight/obese children, aged 7–12 years, daily consumption of 8 g oligofructose-enriched inulin for 16 weeks significantly reduced body weight z-score (reduced by 3.1%), percent body fat (2.4% reduction), and percent trunk fat (3.8% reduction) compared to children who received the placebo who experienced a slight increase in all 3 parameters [214]. The prebiotic group also showed a significant decrease in IL-6 levels from baseline (15% lower), while the placebo group showed an increase (by 25%). Serum triglycerides were also significantly reduced (by 19%) in the prebiotic group. Gut microbiota analysis revealed significant increases in *Bifidobacterium* species and decreases in *Bacteroides vulgatus* in the prebiotic group. Levels of primary bile acids increased in the placebo group but remained unchanged in the prebiotic group over the 16-week period. However, twelve weeks of oligofructose consumption at the same quantity in obese and overweight children, aged 7–11 years (8 g prebiotic/day) and aged 12–18 years (15 g/day) in a double-blind placebo-controlled trial had no impact on body weight and body fat [215].

Consumption of inulin-type fructans by obese women at 16 g/day for 3 months led to gut microbiota changes which included an increase in *Bifidobacterium* and *F. prausnitzii*, both of which negatively correlated with serum lipopolysaccharides [216] The prebiotic also decreased *Bacteroides intestinalis*, *Bac. vulgatus* and *Propionibacterium* which was associated with a slight decrease in fat mass and with phosphatidylcholine and plasma lactate levels. The authors suggest that the modest changes in host metabolism indicate a role for inulin-type fructans to support dietary advice with regards obesity and related metabolic disorders. In a later randomized, double-blind, parallel, placebo controlled trial, obese women consuming the same prebiotic at the same concentration for three months had significantly lower total SCFAs, acetate and propionate (that positively correlated with BMI), as well as significantly lower fasting insulinemia and homeostasis model assessment (indicator of insulin resistance) compared to the placebo group [217]. The following species were significantly increased in the prebiotic group at the end of the three months, *Bifidobacterium adolescentis*, *Bifidobacterium pseudocatenulatum* and *Bif. longum*, the latter of which negatively correlated with serum lipopolysaccharide and endotoxin. 

Synbiotics have shown some promise towards improving growth outcomes in healthy and malnourished children, although there is a paucity of clinical trials in this area. For example, Malawian children, aged 5 to 168 months, suffering from SAM who received ready-to-use-therapeutic-food (RUTF) with a synbiotic for approximately 33 days (median) in a double-blind efficacy randomized controlled trial showed a trend towards reduced outpatient mortality when compared to those who received RUTF alone (*P* = 0.06) [218]. Despite this, the study showed no differences between both groups in terms of nutritional cure, weight gain, time to cure, and prevalence of clinical symptoms including respiratory issues, fever and diarrhoea. One year consumption of a probiotic- (*Bif. lactis*, 1.9 × 10^10^ cfu/day) and prebiotic-fortified milk by Indian preschool healthy and stunted children resulted in increased weight gain (0.13 kg/year, *P* = 0.02) and reduced risk of being anemic and iron deficient (*P* = 0.01) compared to children receiving control milk [219]. A synbiotic consisting of *Bif. longum*, *L. rhamnosus* and inulin and FOS fed to healthy 12-month-old toddlers in milk for one year significantly improved weight gain compared to those receiving control milk (difference of 0.93 g/day) [220]. The weight gain resulted in a change in z-score weight-for-age closer to the WHO Child Growth Standard. Fecal lactobacilli and enterococcal counts were also significantly increased in the synbiotic group between 12 and 16 months. A six-month synbiotic supplementation to children with failure to thrive, a common problem in children in underdeveloped countries, resulted in a significant increase in weight gain compared to control children [221]. Indeed, by the end of the six-month trial, the mean weight of the control group was 11.760 ± 0.17 kg which increased from 10.75 ± 0.16 kg initially, while the mean weight of the synbiotic group was 12.280 ± 0.190 kg, increasing from 10.25 ± 0.2 kg initially. More clinical trials investigating the impact of combinations of probiotics and prebiotics on undernutrition are required. An emphasis on gut microbiota changes should help to delineate mode of action and identify the most suitable formulations for specific conditions. The studies discussed in this section are summarized in Table 3.

### 3.3. Fibre

In order to fully appreciate the impact of fibre on the gut microbiota it is important to be aware of the different types and their properties and in this respect several classification systems have been proposed. That proposed by Ha et al. [13] classifies fibres into those that are “microbially degradable” and those that are “microbially undegradable.” Combining microbial degradability with the other main properties of fibre, including viscosity and water solubility, Bozzetto et al. [222] presented four main groups based on the concepts of McRorie et al. [223]: (1) non-viscous, insoluble, non-fermentable fibre, e.g., bran, cellulose, hemicelluloses, lignin; (2) non-viscous, soluble, fermentable fibre, e.g., inulin, dextrin, oligosaccharides, resistant starch; (3) viscous, soluble, fermentable fibre, e.g., pectin, β-glucan, guar gum and glucomannan; (4) viscous, soluble, non-fermentable fibre, e.g., psyllium, methylcellulose. Different fibres can exert different effects on the gut microbiota and hence have different physiological consequences for the host (Table 4).

In humans, increased fibre intake has been shown to improve certain metabolic parameters associated with obesity and its co-morbidities, such as serum cholesterol levels, particularly in conjunction with energy-controlled dietary regimes. For example, in overweight and obese adults (BMI = 25 to 45), daily consumption of two portions of whole-grain ready-to-eat oat cereal (3 g/day oat β-glucan) as part of a reduced energy dietary programme (~500 kcal/day deficit) with regular physical activity for 12 weeks proved more effective than an energy-matched low fibre diet for reducing LDL cholesterol levels (*P* = 0.005), total cholesterol (*P* = 0.038), and non-HDL cholesterol (*P* = 0.046) [224]. While weight loss did not differ between groups, there was a significant difference in waist circumference as a result of eating the high fibre diet, resulting in a loss of ~3.3 cm versus only ~1.9 cm on the low fibre diet (*P* = 0.012). Daily consumption of whole grain wheat bread by Japanese subjects (BMI ≥ 23) for 12 weeks resulted in a significant reduction in visceral fat area (−4 cm^2^) which was not observed in subjects consuming refined white bread [225]. Similarly, whole grain wheat consumption in conjunction with an energy restricted diet for 12 weeks by post-menopausal women resulted in a greater reduction in body fat percentage (−3.0%) compared to consumption of refined wheat (−2.1%) [226]. While serum total and LDL cholesterol increased by ~5% in the refined wheat group (*P* < 0.01), they did not change in the whole wheat group. Body weight decreased significantly for both groups but did not differ between groups. Consumption of the recommended intake levels of dietary fibre and fat in obese and overweight (BMI = 30.7) pregnant women positively associated with gut microbiota richness whereas high fat with low fibre and low carbohydrate consumption associated with significantly lower gut microbiota richness [227]. The richer gut microbiota correlated with lower maternal inflammatory status. In another study involving overweight and obese pregnant women, low fibre intake was found to increase the genus *Collinsella* in the gut microbiota, which is positively associated with circulating insulin [228]. The low fibre diet was also associated with a gut microbiota favouring lactate fermentation, whereas the high fibre diet was associated with SCFA producing bacteria.

In a study investigating the existence of a correlation between body weight change over time and gut microbiome composition involving 1632 healthy females from TwinsUK, (national register of adult twins for studying age-related complex traits and disease), Menni et al. [229] found that gut microbiota diversity was negatively associated with long-term weight gain, but positively associated with fibre intake independent of calorie intake or other confounders.

These studies indicate that dietary fibre has a role to play in the control of obesity and its related comorbidities. Further research is needed in order to fully comprehend the impact of the different dietary fibres on the gut microbiota and to delineate the subsequent consequences for host metabolic health. However, the human microbiota is characterised by extensive inter-individual variation, with genetics a significant contributing factor, and it is now becoming clear that the composition of an individual’s microbiota will determine how it responds to dietary components, in particular fibre. In this regard, understanding what ‘responding’ and ‘non-responding’ microbiota look like is essential as well as how to convert a ‘non-responder’ into a ‘responder.’

## 4. The Microbiota Can Be Used as a Biomarker to Predict Responsiveness to Specific Dietary Consitituents, For Example, Fibre

We know that diet can directly or indirectly influence the gut microbiota, and studies have shown that this response can be rapid with changes observed within 1–3 days [7,230,231] when the modifications are “large” [232] such as the all-animal or all-plant products diet [7], or large increases/decreases in fibre [230,231]. However, inter-individual variance is often much greater than the variance introduced as a result of diet [233]. Indeed, an individual’s baseline microbiota and health status at the beginning of an intervention influences the extent of potential changes to the microbiota and subsequently the host, and studies are showing that baseline microbiota consisting of responders and non-responders to dietary interventions as well as effectors of host responses (both of which may be the same microorganism or a consortia of microorganisms), is generally linked to habitual dietary trends [6,230,234]. In this regard, an individual’s baseline microbiota harbors predictive potential with regard to the effect of dietary constituents on the host and this has been proven particularly in case the of fibre.

Salonen et al. [233] reported that high microbiota diversity before dietary intervention with resistant starch or non-starch polysaccharides associated to low dietary responsiveness of the microbiota. Similarly, obese individuals with low microbial gene richness (low microbiota diversity) in their initial faecal microbiota showed a greater microbiota response in terms of gene richness to a weight loss diet compared to obese individuals represented by high microbial gene richness in their initial microbiota [234]. However, individuals with high gene richness showed a more marked improvement in systemic inflammation and adipose tissue following the intervention suggesting that gene richness could provide a predictive tool towards intervention efficacy in relation to inflammatory variables. Tap et al. [235] also reported that low OTU microbiota richness was associated with a greater microbiota change over time following a large increase in dietary fibre (40 g/day) in healthy adults for 6 weeks whereas high OTU microbiota richness at baseline proved more stable upon high dietary fibre intervention and was associated with high proportions of *Prevotella* and *Coprococcus* species and a higher *Prevotella*:*Bacteroides* ratio.

Indeed, a number of studies have reported associations between the abundances or lack of abundance of specific species and the responsiveness of the microbiota and the host to dietary intervention. Two overweight men who failed to ferment significant amounts of resistant starch during a 10-week intervention involving a total of 14 participants showed very low numbers of R-ruminococci (relatives of *Ruminococcus bromii*) and were also non-methanogenic [231]. The gut microbiota of healthy subjects exhibiting improved glucose tolerance following three days of consumption of barley kernel-based bread was enriched with *Prevotella copri* and after the intervention exhibited a higher *Prevotella*:*Bacteroides* ratio compared to non-responders [236]. However, in a follow-on study, the researchers failed to stratify metabolic responders and non-responders based on *Prevotella* and *Bacteroides* abundance at baseline, but those with the highest *Prevotella*:*Bacteroides* ratio at the beginning of the study displayed improved appetite sensations (less hunger and less desire to eat), reduced insulin responses and reduced inflammatory markers compared to those with the lowest *Prevotella*:*Bacteroides* ratio independent of the intervention suggesting that the higher *Prevotella*:*Bacteroides* ratio is favourable [237]. Oligotyping of 16s rRNA gene sequencing data, which permits resolution to species level and below, enabled De Filippis et al. [238] to identify distinctive correlation patterns between *Prevotella* and *Bacteroides* oligotypes with dietary components and metabolome using faecal samples from omnivore and non-omnivore subjects. The authors concluded that an indiscriminate association between a whole genus and a specific dietary pattern may result in an oversimplified vision of correlations between gut microbiota and diet, failing to take diversity within a genus or even a species into account. Based on three independent cohorts of obese adults from Finland [239], Belgium [216] and Britain [231] involved in different dietary interventions (fibre/prebiotics/weight loss diet) to improve metabolic health, Korpela et al. [240] reported that baseline microbiota of non-responders (in terms of gut microbiota changes) was characterised by average abundances of two Firmicutes species, *Eubacterium ruminantium* and *Clostridium felsineum,* which were present at very low or very high baseline abundances in responders. Furthermore, the presence of high levels of *Clostridium sphenoides*, a common gut inhabitant and Firmicutes member, in the faecal microbiota of obese individuals before dietary invention was associated with a decrease in cholesterol following intervention while obese individuals with abnormally low abundance of this species did not benefit in terms of cholesterol levels. Interestingly, *C. sphenoides* abundance was not associated with absolute levels of cholesterol and so may not be directly involved in cholesterol metabolism. Dietary fibres were shown to promote a select group of SCFA-producing bacteria in patients with type 2 diabetes [241]. However, when present at greater abundance and diversity, the authors reported an improvement in haemoglobin A1c levels (glycosylated haemoglobin), partly due to increased glucagon-like peptide (GLP)-1 production, and diminution of producers of metabolically detrimental compounds. In a randomized controlled trial investigating the impact of increased intake of whole grains versus fruits and vegetables on the gut microbiota in obese and overweight individuals, Kopf et al. [242] reported that both treatments induced individualised changes but that baseline levels of Clostridiales correlated with the magnitude of change in lipopolysaccharide binding protein which is indicative of change in inflammatory state.

The influence of long-term dietary habits, in particular habitual fibre intake, on gut microbiota responsiveness to specific interventions is now becoming apparent. In a randomized, double-blind, placebo-controlled, cross-over study, Healey et al. [243] classified participants as either high or low dietary fibre consumers prior to three weeks of daily supplementation with an inulin-type fructan prebiotic. The high dietary fibre group revealed significant increases in the relative abundances of *Bifidobacterium* and *Faecalibacterium* along with significant reductions in *Coprococcus*, *Dorea* and *Ruminococcus* (*Lachnospiraceae* family). The gut microbiota of the low dietary fibre group was less responsive showing only an increase in *Bifidobacterium*. Based on an *in vitro* approach, Brahma et al. [244] investigated the impact of donor dietary pattern on the fermentation properties of whole grains and brans. Although the samples were taken from donors with similar energy intakes, they differed in terms of their intakes of several beneficial nutrients. Samples from G1 subjects were representative of the superior diet while samples from G2 subjects represented the inferior diet. The G1 microbiota showed higher diversity and greater abundances of beneficial microbes including *Faecalibacterium* and was better equipped to metabolise the complex carbohydrates than the microbiota from G2 subjects, resulting in greater butyrate production, while the microbiota of G2 subjects produced more acetate and propionate. In another study, Griffin et al. [245] reported that Americans consuming unrestricted diets had less diverse faecal microbiota (termed AMER) compared to the microbiota of individuals consuming calorie-restricted plant-rich diets (termed CRON) and the AMER microbiota lacked many bacterial lineages representative of CRON. Interestingly, transplanting AMER microbiota into gnotobiotic mice and feeding them the CRON diet resulted in community configurations but which were weaker than their CRON counterparts. Placing the AMER communities into a model meta-community composed of several CRON communities resulted in the dispersal of microorganisms between the coprophagic animals which enhanced the reconfiguration of the AMER microbiota in response to the CRON diet and resulted in changes in host metabolic features, all driven by an influx of CRON dietary practice-associated taxa. This artificial metacommunity model provides an opportunity to mine multiple human microbiota for microbial reporters of responses to diet as well as effectors of host response. However, Sonnenburg et al. [246] showed that while microbiota changes in mice resulting from a low MAC diet could be reversed within a single generation by reintroducing MACs, the progressive loss in diversity over multiple generations consuming the low MAC diet could not be reversed with the reintroduction of dietary MACs alone. Importantly, restoration of the microbiota to its original state required the reintroduction of lost taxa along with dietary MACs.

These studies indicate that the microbiota has the potential to serve as an effective biomarker to predict responsiveness to specific dietary constituents with most if not all studies to date focusing on fibre/complex carbohydrates. The responsiveness of the gut microbiota (including responders and effectors of host responses) appears to be largely dependent on baseline microbiota diversity and the specific microbes present or absent at baseline, the latter of which can have a profound influence on the poorly diverse microbiota. Indeed, a highly diverse microbiota as a result of long-term, healthy dietary practices involving adequate fibre consumption remains stable in the face of fibre intervention, is rich in both responders and effectors and is capable of reaping the metabolic benefits for the host. A gut microbiota with low diversity can benefit from dietary intervention but only if the specific responder and effector microbes are actually present even at low abundances. Indeed, Healey et al. [243] showed that lower baseline bifidobacteria concentrations in subjects correlated with a more pronounced bifidogenic response following prebiotic intervention. But poor dietary practices and insufficient dietary fibre intake over a long-term period may actually result in the extinction of beneficial microbial lineages. In this case, dietary intervention with fibre/prebiotics will fail to mitigate a beneficial outcome for the host and will presumably require the addition of specific taxa along with their corresponding MACs in the form of synbiotics. However, despite the presence of resistant-starch degrading microorganisms at low abundances in a subset of healthy young adults, the consumption of resistant starch failed to increase their abundances [247]. Such a phenomenon may be due to the presence of antagonistic microorganisms to the resistant starch-degrading microbes which the authors suggest could require targeted removal prior to intervention and could include the presence of bacteriophages. Another form of dietary fibre may be more suited to the particular microbiota in these individuals, or the synbiotic approach may be required. Clearly, more studies are required to determine gut microbiota responses to specific dietary components, i.e. targeted microbiota dietary intervention, using a top-down approach of gut microbiota analysis from diversity levels to species and even strains, to their gene content and functionality (metabolome, transcriptome, proteome), alongside host clinical and genetic data, for input into machine learning algorithms designed to identify correlations, which subsequently can be investigated for causal evidence, in order to accurately predict individualised responses for maximized health (Figure 1). Indeed, machine learning models for predicting disease from metagenomic datasets have already been developed [248]. Thus, already the potential of the gut microbiota as a biomarker of responsiveness to diet is tangible with opportunity for precision microbiomics beginning to emerge. However, ‘causal evidence,’ is a critical factor in this workflow and in Section 7 we provide guidelines for evaluating the scientific validity of evidence for providing personalised microbiome-based dietary advice.

## 5. Opportunities for Precision Microbiomics

Understanding how the microbiome responds to dietary constituents and the subsequent clinical consequences for the host can be used in the design of precision-tailored diets which ensure maximal nutritional/functional outcome for the host. However, to date only a handful of studies are available to provide specific examples of precision microbiomics in nutrition. For example, composition and functional alterations observed in the faecal metagenome of 145 European women with type 2 diabetes were integrated into a mathematical model that enabled accurate prediction of type 2 diabetes based on metagenomic profiles [249]. The model was capable of identifying women with a diabetes-like metabolism among a group with impaired glucose tolerance. However, the model failed to work on a Chinese cohort revealing that discriminant metagenomic markers for type 2 diabetes differ between Chinese and European cohorts and should be age- and geography-specific. Another example is the direct modulation of colonic microbiota with short-chain GOS to metabolize lactose in lactose intolerant individuals [250]. In this case, GOS failed to elicit a bifidogenic response in three out of 30 participants; however, an increase in bifidobacteria was associated with a decrease in pain and cramping revealing its significance in terms of symptoms. Cho et al. [251] reported that high TMAO producers amongst healthy male adults (≥ 20% increase in urinary TMAO in response to beef and eggs) had significantly more Firmicutes than Bacteroidetes and significantly less microbiota diversity. While the results are based on a short-term feeding study, longer-term feeding trials involving larger cohorts coupled with microbiome data could enable accurate prediction of a high TMAO-producing microbiota and subsequent strategies to alter it. Maintaining normal blood glucose levels is critical for the prevention and control of metabolic syndrome [252] but blood glucose levels are rising at an increased rate as evidenced by the prevalence of prediabetes and impaired glucose tolerance in the general population [253]. Food choices that induce normal PPGRs are critical for controlling blood glucose levels which are in essence controlled by dietary intake. However, until recently, no method existed to predict PPGRs to food. Over the period of a week Zeevi et al. [14] continuously monitored PPGRs in a cohort of 800 healthy and prediabetic individuals in Israel in response to identical meals and noted high variation. They also measured physical activity, anthropometrics, blood parameters, gut microbiota composition and function, as well as self-reported lifestyle behaviours. This multidimensional data was integrated into a machine learning algorithm that was capable of accurately predicting personalised PPGRs and was further validated in an independent cohort of 100 people. Interestingly, the highly variable PPGRs in individuals associated with multiple person-specific microbiome and clinical factors, and tailored diets based on predictions from a machine learning algorithm not only significantly improved PPGRs but also resulted in consistent alterations to the gut microbiota. This study was recently replicated in a different population (from the USA) [254]. Based on a randomized cross-over trial involving 20 healthy subjects comparing the effects of consuming either traditionally made sourdough leavened whole-grain bread or industrially made white bread for one week each, Korem et al. [255] found that the glycemic response varied significantly in response to the different bread types and the type of bread which induced the higher glycemic response in each person could be predicted using microbiome data recorded just prior to the intervention. However, the exact mechanisms involved in the gut microbiota and glycemic control remain to be elucidated.

Non-calorific artificial sweeteners (NAS) were developed to provide sweet taste to foods without the high-energy content of calorie-rich sugars. However, Suez et al. [256] reported that long-term consumption of commonly used NAS in humans significantly and positively correlated with several metabolic syndrome-related clinical parameters including measures of central adiposity, higher fasting blood glucose, higher haemoglobin A1c%, and higher measures of impaired glucose tolerance. Moreover, statistically significant positive correlations were found between multiple taxonomic entities and long term NAS consumption including the *Enterobacteriaceae* family, the Deltaproteobacteria class and the Actinobacteria phylum. In order to determine if the relationship between blood glucose control and NAS consumption was causal, Suez et al. [256] followed seven healthy volunteers (who did not normally consume NAS in any form) who consumed the FDAs maximum acceptable daily intake of saccharin (5 mg/kg body weight) for 5 days. Four of the seven individuals developed significantly poorer glycemic responses 5–7 days after NAS consumption. Interestingly, the microbiome of NAS responders was distinct from NAS non-responders both before and after NAS consumption and the microbiome of NAS non-responders featured minimal changes after the NAS intervention in contrast to the pronounced compositional changes observed in NAS responders. Transferring day 7 stools from NAS responders into normal germ-free mice induced significant glucose intolerance compared to mice transplanted with day 1 stool (before intervention) from the same NAS responders. Similarly, day 7 stools from non-NAS responders induced normal glucose tolerance in mice. Furthermore, germ-free mice transplanted with responders’ day 7 stool replicated some of the dysbiosis observed in humans including a 20-fold increase in *Bacteroides fragilis* (order Bacteroides) and *Weissella cibaria* (order Lactobacillales) and a 10-fold decrease in *Candidatus* Arthromitus (order Clostridiales) [256]; this over-representation of Bacteroides and underrepresentation of Clostridiales has been previously associated with type 2 diabetes in humans [249,257]. Thus, humans exhibit a personalised response to NAS as a result of their microbiota composition and functionality which as the authors state strongly suggests that other nutritional responses may be driven by “personalised functional differences in the microbiome,” and the resulting opportunity for “personalised nutrition” may lead to “personalised medical outcome.” Wang et al. [258] recently described the bacteriostatic effect of non-nutritive sweeteners such as sucralose and stevia in mice.

## 6. Commercialisation of Microbiome Testing

Gut microbiome testing is currently commercially available and takes advantage of the reduced costs associated with next-generation sequencing technologies. Table 5 provides a non-exhaustive list of these companies. While several companies provide doctor-ordered stool tests (e.g., the SmartGut test provided by Ubiome and Genova Diagnostics, USA, etc.) the majority of the companies listed in Table 5 provide direct-to-consumer tests.

The sequencing method used to analyse the gut microbiome has a significant impact on cost, given that companies providing 16s rRNA gene sequencing are generally cheaper (approximately $100/test) than those that use whole genome sequencing and metatranscriptomics (approximately $350 to $400/test). However, the latter two also provide information regarding the metabolic potential of the gut microbiome, providing insights into microbiota-derived metabolites related to health and disease.

Regulation of commercial microbiome testing in specific markets remains unclear and the need for a clear global regulatory direction and guideline is required to advance testing and thus its impact on human health. Furthermore, some commercial laboratories will often modify/optimise their sequencing methods which can lead to inconsistencies when comparing results from different companies. Of course, this has potential risks with regards to interpretation and transferability of results, highlighting the need to develop a set of guidelines to assure consistency in the way different laboratories operate. The Microbiome Quality Control project (MBQC) has been set up to govern such guidelines (https://www.mbqc.org/).

Many companies provide easy-to-understand, detailed reports regarding gut microbiota diversity, microbial members including beneficial and pathogenic microorganisms which influence health and disease, a comparison of the individual’s gut microbiome to other participants, and personalised dietary, supplemental and lifestyle advice. Importantly, such tests are not diagnostic given the current level of evidence that is available regarding the gut microbiome and it is essential that consumers availing of these tests are aware of this fact and seek medical advice if experiencing symptoms of any kind rather than self-diagnosing and self-healing via the provided advice which at most can only serve as a personalised guideline. Indeed, many medical professionals and microbiome experts remain dubious about the utility of these direct-to-consumer tests due to the lack of concrete evidence linking particular microbiota signatures to specific host phenotypes including disease, disease risk and potential treatment responses. Over-extrapolation of results on the side of the service provider and over-interpretation of results on the side of the consumer are also risk factors. Indeed, over-interpretation of results on the side of the consumer may lead to unnecessary anxiety and the adaptation of dietary alterations as well as intake of supplements which may do more harm than good or have no effect at all. In addition, we have already mentioned that whole genome sequencing is more informative than 16s rRNA gene sequencing and this is something the consumer should be made aware of. For example, subgroups A, B and C of *F. prausnitzii* are not discriminated from each other with 16S rRNA gene sequencing, but are identified by metagenomic and metatranscriptomic sequencing.

Most commercially available 16S rRNA-based tests report on the relative abundance of *F. prausnitzii* with no differentiation between *F. prausnitzii* subgroups A, B and C. This may lead to a misleading interpretation of results as it has recently been found that different subgroups produce butyrate at different levels and have been linked to different disease states. For example, *F. prausnitzii* A produces comparatively lower levels of butyrate and at high levels has been linked to colon cancer, appendicitis and inflammatory conditions. Similarly, *F. prausnitzii* B also produces comparatively lower levels of butyrate and at high levels has been linked to atopic dermatitis. Conversely, *F. prausnitzii* C has been shown to produce the highest level of butyrate of all the three subgroups and also produces an anti-inflammatory protein called MAM. As a result, higher levels of *F. prausnitzii* C are thought to be anti-inflammatory whereas low levels have been linked to Crohn’s disease, ulcerative colitis, colon cancer, type II diabetes and chronic fatigue syndrome [259]. This highlights the importance of understanding the relative abundance of *F. prausnitzii* subgroups A, B, and C when drawing conclusions about butyrate production and association with disease.

Furthermore, in terms of time, the tests can take anything from two to eight weeks before the consumer receives the results such that gut microbiota changes may have taken place within this time frame depending on the consumer’s circumstances (e.g., dietary changes, medical treatment, antibiotic administration, etc) and hence the results may prove meaningless by the time of receipt, a factor the consumer must be made aware of. Indeed, regular microbiome testing would prove more effective but, at this moment in time, may prove cost-prohibitive for most consumers. Despite this, some companies offer discounts for regular microbiome testing and the results of such tests will provide essential data regarding the impact of personalised nutritional advice (assuming the consumer follows it) on the gut microbiome along with its long-term effects. While the analysis is capable of providing an insight into the necessary dietary recommendations to achieve a ‘healthy’ gut microbiome, one must question the utility of this information at present given that we have yet to define a universal ‘healthy’ gut microbiome, which may not be possible given the suspected level of specificity that can be associated with an individual’s ‘healthy’ gut. Indeed, a food questionnaire would generate sufficient information to provide personalised dietary recommendations which should subsequently improve the status of the gut microbiome. However, in its favour, gut microbiome testing represents a useful tool in its present state to increase awareness of the gut microbiome and its influence on overall health and the more tests that are performed, the greater the opportunity to move towards precision microbiomics by advancing our current knowledge base. 

In relation to future testing, it is also important to consider the importance of host genetics/gene expression, and considering how host genetics can impact the gut microbiome, and be used as proxy for providing personalised dietary advice. For example, numerous genetic variations have been linked to influencing a range of microbiota [260,261] as well as beta-diversity [260]. However, other factors such as diet may mask the effect of genetics on the microbiome making it difficult to predict changes in phenotype without assessing an individual’s diet and including this in the interpretation. An example whereby the assessment of host genetics in a commercial setting shows utility in providing personalised microbiome-based recommendations is the association between FUT2 genotype/secretor status and the expression of fucosyllated glycans on host cell surfaces and in secretions [262]. Common FUT2 polymorphisms have been shown to influence the expression of fucosyltransferase2, an important enzyme associated with the production of the dominant human milk oligosaccharide, 2’-fucosyllactose (2’FL), and other fucosyllated oligosaccharides. Lactating mothers who possess the inactive form of the FUT2 polymorphism (approximately 20% of the Caucasian population) do not contain 2’-FL in their breast milk. The absence of this gene (non-secretors) has been associated with delayed establishment of *Bifidobacterium* spp. in the infant gut and increased risk of diabetes, alcohol-induced pancreatitis and Crohns disease. Interestingly, non-secretor status has also been associated with resistance to infectious disease such as norovirus and rotavirus infection and *Helicobacter pylori* colonization. As such, genotyping for FUT2 secretor status allows for the identification of infants and adults that can benefit from treatment with probiotics, prebiotics and other dietary components. Hence, future commercial tests may offer genetic testing as a way to help consumers, such as lactating mothers in the case of FUT2, make better choices and optimise health outcomes.

## 7. Guidelines for Evaluating the Scientific Validity of Evidence for Providing Personalised Microbiome-Based Dietary Advice

As noted in the previous section, there are risks associated with the rapid commercialisation of microbiome testing including inconsistencies in results between laboratories as well as over-extrapolation of results on the side of the service provider and over-interpretation of results on the side of the consumer. To mitigate such risks, guidelines for evaluating the scientific validity of evidence for providing personalised microbiome-based dietary advice need to be developed. A similar set of guidelines has been proposed for genotype-based dietary advice [263] providing a useful template from which to start.

The guidelines proposed by Grimaldi et al. [263] provide a framework for assessing the strength of the evidence and scientific validity of gene(s) × diet interactions which help determine the ‘actionability’ of the interaction. Such guidelines can be modified and applied to precision nutrition and the microbiome. These guidelines use the ACCE model (Analytical and Clinical Validity, Clinical Utility and Ethics) as the starting point according to which a medical genetic test should fulfil requirements regarding:
Analytical validity—a measure of the accuracy of the genotyping.Scientific validity—concerns the strength of the evidence linking a genetic variant with a specific outcome.Clinical utility—the measure of the likelihood that the recommended advice or therapy will lead to a beneficial outcome beyond the current state of the art.Ethical, legal and social implications that may arise in the context of using the test.

For conducting an assessment according to precision nutrition and the microbiome, analytical validity should be relatively straightforward as projects such as MBQC have been set up to assure consistency by applying standard operating procedures and best practices in how laboratories operate in the microbiome testing field. Similarly, the requirements for scientific validity could also be fulfilled, whereby scientific validity in the context of precision nutrition and the microbiome refers to the strength of the evidence for an interaction between a specific microbiome biomarker or a microbial enterotype and a dietary component or a specific health outcome, disease or risk factors for disease.

The requirements for clinical utility, on the other hand, may be harder to fulfil as it has strict criteria in the medical sense, demanding strong evidence that a given therapy ‘will lead to an improved health outcome’ [264,265]. A caveat is that defining an ‘improved health outcome’ due to microbiome-based advice in a generally healthy person is very hard. Additionally, we are still unsure as to what constitutes a ’healthy‘ microbiome, making it even harder to define an ‘improved health outcome’. With regard to the ethical, legal and social implications, as with personalised nutrition, existing rules must be developed for microbiome testing to ensure that the fundamental rights of the consumer are protected and legislation should be put in place to identify direct-to-consumer tests which provide non-scientifically validated information and advice [264,265]. Ethical, legal and social implications of the human microbiome have been discussed elsewhere [266,267]. Therefore, as per the framework proposed by Grimaldi et al. [263], a guideline that evaluates the scientific validity and evidence for providing personalised microbiome-based dietary advice should focus primarily on the assessment of scientific validity, an essential requirement before any nutrition advice should be given.

### Proposed Framework for Scientific Evidence Assessment

The scientific validity assessment criteria for microbiota-based dietary advice within an adapted framework would include (i) study design and quality, (ii) biological mechanism and plausibility and (iii) the probability term (Table 6). Whilst the assessment of study design and quality as well as biological mechanisms and plausibility are commonly used to assess the value of scientific evidence, the use of a ‘probability term’ is not as common. The probability term is the overall judgement of the evidence provided and is based on the European Food Safety Authority (EFSA) guidance document expressing uncertainty in scientific assessment [268] to help describe the likelihood of an outcome where firm conclusions are difficult to draw. A probability term makes it possible for an ‘evidentiary conclusion based on many papers, each of which may be relatively weak, to be graded as ‘moderate’ [probable] or even ‘strong’ [convincing], if there are multiple small case reports or studies that are all supportive with no contradictory studies”. For more detail on the definition and use of each probability term, study design and quality, biological plausibility as well as examples of the type criteria that can be used to assess gene/microbiome x diet interactions, please refer to Grimaldi et al. [263].

The fundamental requirement of a nutrition test (genetic, metabolites, microbiota), as with any health-related test, is that the results should indicate clearly a diet-related recommendation that is beneficial in relation to a concrete aspect of health or performance. Any such advice should fulfil all requirements set out in the framework described here. Inevitably, any assessment of nutrition can only be semi-quantitative at best. We consider that the approach of this framework has the benefit of creating a formal and generic model for the assessment of such evidence and will guide more focused debates on specific points, which may be judged in different ways. Moreover, the framework and associated resources will allow stakeholders such as dietitians, nutritionists and genetic counsellors to improve their knowledge of the microbiome and, at the same time, will provide a valuable resource to assess the various tests that are offered. This framework may also encourage a greater standardisation of research protocols, supporting other initiatives, as well as the reporting of novel and replicated microbiome-environment interactions in other populations.

## 8. Conclusions

The field of gut microbiota research boasts thousands of studies, the majority of which have been published in recent years. Many of these are observational, documenting differences between healthy and diseased states allowing for correlations to be made between diversity scores, specific taxa, disease, disease risk and health status, and have been essential in our understanding of the significance of the gut microbiota to overall health and disease. Interventions have highlighted the significance of inter-individual variation in terms of intervention efficacy and this is most apparent for fibre, presumably due to the ability to measure the expected outcome i.e. modulation of gut microbiota composition, increases in SCFAs. Thus, understanding why and how an intervention has failed at the individual level is as critical as understanding why and how it has succeeded. With this in mind, it seems the field is on the brink of being propelled towards precision microbiomics, where inter-individual variation is being embraced and correlation studies are beginning to be supported by causal evidence through thorough experimental validation. This will allow for the design of strategic interventions and ultimately evidence-based dietary advice at the individual level. Criteria ensuring scientific validity for microbiota-based dietary advice are, thus, critical. Such data will not only serve nutrition counselling but will also prove valuable in the field of medicine for the clinical/therapeutic management of individuals. Furthermore, other members of the microbiota including the phageome, virome and mycobiome, are likely to contribute to human health as much as the bacterial component and should be included in analysis in order to gain comprehensive insight. Indeed, precision nutrition through the microbiome offers individuals huge potential to manage disease risk through diet and microbiome-modulating interventions and thus improve both quality and longevity of life.

## Figures and Tables

**Figure 1 nutrients-11-01468-f001:**
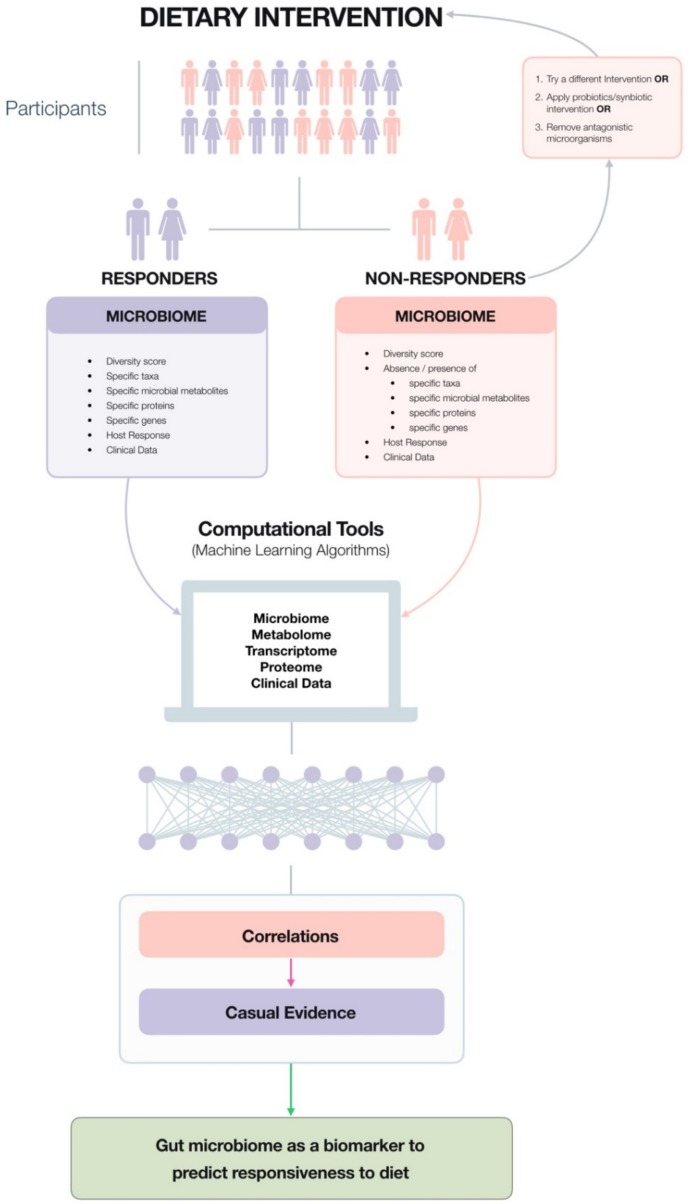
Diagrammatic representation of the sequence of events involved in accurately predicting individualised responses to diet and options for converting a non-responder into a responder. (graphic design for Figure 1 by Chor Hung Tsang, Mad Lemur Design Studio).

**Table 1 nutrients-11-01468-t001:** Potential benefits of probiotics, prebiotics, synbiotics and fibre on different life stages.

Life Stage	Probiotics	Prebiotics/Oligosaccharides	Synbiotics	Fibre
Pregnancy/Lactating Mother	— Reduce incidence of ^a^GDM [37] — Reduce risk of preeclampsia [40,41,42] — Lower risk of preterm delivery [42] — Prevent infectious mastitis [44] — Reduce incidence of bacterial vaginosis [43] — Alter immunologic composition of breast milk [56,57]	— Reduce abundance of faecal *Bacteroides* in women with GDM [34] — Alter immunologic composition of breast milk [60]	— Decrease serum insulin levels and positively influence other insulin actions [35] — Reduce blood pressure in women with GDM [36] — Alter immunologic composition of breast milk [61] — Positively impact mineral levels in breast milk [62]	— Reduce abundance of faecal *Bacteroides* in women with GDM [34]
Infant	— Reduce risk of all-cause mortality, ^b^NEC ≥ Stage II, late onset sepsis and feeding intolerance in preterm infant [69] — Increase faecal bifidobacteria, reduce Proteobacteria and Clostridia in breast-fed caesarean born infants [84] — Treat infant colic in breast fed infants [88]	— Decrease incidence of mortality, sepsis, hospital stay duration and time to full enteral feeding in preterm infant [78] — Soften infant stool in formula-fed healthy infants [89]	— Reduce incidence of NEC in preterm infant [80,81,82] — Reduce incidence of sepsis in preterm infants [81] — Increase stool frequency in formula-fed infants [90] — Beneficially modulate gut microbiota, reduce incidence of ear infections and use of dermatological medication in infants with non-^c^IgE-mediated cow’s milk allergy on amino acid based formula [91] — Reduce infantile crying and colic, functional constipation and daily regurgitation in infants on starter formula [92] — Reduce cumulative incidence of lower respiratory tract infections in weaned infants [93] — Beneficially modulate gut microbiota in caesarean born infants [94]	
Adult–Physical Activity	— Improve immunity [147,151,153,161,164] — Reduce incidence of ^d^URTI [147,151,159,164] — Relieve morbidity and symptoms of URTI [153] — Reduce severity of gastrointestinal illness and reduce load of lower respiratory illness symptoms in male competitive cyclists [156] — Reduce number of athletes experiencing URTI and reduce number of illness days [158] — Reduce duration of URTI [160] — Decrease fatigue accumulation during consecutive high intensity exercise [153] — Reduce exercise-induced tryptophan degradation rates [159] — Improve endurance performance [165] — Alleviate oxidative stress, increase plasma-branched amino acids and elevate exercise performance [166]		— Reduce plasma endotoxin unit levels, maintain intestinal permeability, reduce gastrointestinal symptoms [167] Note: Study used probiotic/prebiotic antioxidant intervention.	
Adult–Stress	— Decrease anxiety symptoms in ^e^CFS patients [189] — Relieve psychological stress in healthy volunteers [191] — Reduce Black Depression Inventory scores in individuals with major depressive disorder [192] — Reduce postpartum depression symptoms [193] — Reduce salivary cortisol and plasma tryptophan levels in healthy medical students at examination time and increase faecal serotonin levels after exams [194] — Reduce rate of subjects experiencing abdominal and cold symptoms and number of days of such symptoms in pre-exam period [194] — Beneficially modulate gut microbiota [189]	— Improve anxiety scores in individuals with ^f^IBS [196,198] — Beneficially modulate gut microbiota [196,198] — Reduce salivary cortisol awakening response in healthy volunteers [197] — Decrease attentional vigilance to negative versus positive information in a dot-probe task [197]		
Elderly	— Improve immunity [121,122] — Beneficially modulate gut microbiota [122,123,124,125,126,127]	— Beneficially modulate gut microbiota [128,129] — Immunomodulation [129] — Improve exhaustion and handgrip [132] — Reduce frailty index levels [133]	— Beneficially modulate gut microbiota, improve innate immunity and reduce total and ^g^LDL-cholesterol [134]	— Improve bowel function, improve nutrient density of diet [119]

^a^GDM, gestational diabetes mellitus; ^b^NEC, necrotizing enterocolitis; ^c^IgE, immunoglobulin E; ^d^URTI, upper respiratory tract infection; ^e^CFS, chronic fatigue syndrome; ^f^IBS, irritable bowel syndrome; ^g^LDL, low density lipoprotein.

**Table 2 nutrients-11-01468-t002:** Impact of probiotics on overnutrition in human intervention studies.

Probiotic	Subject Information	Duration	Effect	Reference
*Bif. breve* B-3	Adult volunteers, ^a^BMI 24 to 30 kg/m^2^	12 weeks	— Lowered fat mass — Improved some blood parameters related to liver function and inflammation	[201]
*Bif. breve* B-3	Pre-obese adults (25 ≤ BMI < 30 kg/m^2^)	12 weeks	— Lowered body fat mass and percent body fat — Decreased triglyceride levels and improved ^b^HDL cholesterol from baseline	[202]
*Bif. animalis* ssp. *lactis*	Overweight and obese adults	6 months	— Controlled body fat mass and reduced weight circumference and food intake — Lowered circulating zonulin levels	[203]
*L. gasseri* BNR17	Systematic review of human studies	-	— Reduced weight gain	[204]

^a^BMI, body mass index; ^b^HDL, high density lipoprotein.

**Table 3 nutrients-11-01468-t003:** Impact of prebiotics and synbiotics on overnutrition and undernutrition in human intervention studies.

Prebiotic/Synbiotic	Subject Information	Duration	Effect	Reference
Oligofructose	Healthy adults, ^a^BMI > 25 kg/m^2^	12 weeks	— Reduced weight gain — Improved glucose regulation — Reduced caloric intake — Suppressed ghrelin expression — Enhanced ^b^PYY expression	[213]
Oligofructose-enriched inulin	Overweight/obese children, 7–12 years	16 weeks	— Modulated gut microbiota — Reduced body weight z-score, percent body fat and percent trunk fat — Decreased ^c^IL-6 levels — Reduced serum triglycerides	[214]
Oligofructose	Obese/overweight children, 7–11 years and 12–18 years	12 weeks	No effect	[215]
Inulin-type fructans	Obese women	3 months	— Modulated gut microbiota — Slight decrease in fat mass and phosphatidylcholine and plasma lactate levels	[216]
Inulin-type fructans	Obese women	3 months	— Modulated gut microbiota — Lowered total ^d^SCFAs, acetate and propionate — Lowered fasting insulinemia and homeostasis model assessment	[217]
Synbiotic in ^e^RUTF	Children with ^f^SAM, 5–168 months	33 days	— Reduced outpatient mortality	[218]
Prebiotic-probiotic fortified milk	Preschool healthy and stunted children	1 year	— Increased weight gain — Reduced risk of anemia and iron deficiency	[219]
Synbiotic (probiotic mix + inulin and ^g^FOS)	Healthy toddlers, 12 months	1 year	— Modulated gut microbiota — Improved weight gain	[220]
Synbiotic	Children with failure to thrive	6 months	— Increased weight gain	[221]

^a^BMI, body mass index; ^b^PYY, peptide YY; ^c^IL-6, interleukin-6; ^d^SCFAs, short chain fatty acids; ^e^RUTF, ready to use therapeutic food; ^f^SAM, severe acute malnutrition; ^g^FOS, fructooligosaccharide.

**Table 4 nutrients-11-01468-t004:** Impact of fibre on overnutrition in human intervention studies.

Fibre	Subject Information	Duration	Effect	Reference
Oat β-glucan	Overweight obese adults, ^a^BMI = 25 to 45 kg/m^2^	12 weeks	— Reduced ^b^LDL cholesterol, total cholesterol and non-^c^HDL cholesterol — Reduced weight circumference	[224]
Whole grain wheat bread	BMI ≥ 23 kg/m^2^	12 weeks	— Reduced visceral fat area	[225]
Whole grain wheat	Post-menopausal women	12 weeks	— Reduced body fat percentage — Controlled serum total and LDL cholesterol	[226]
Recommended intake of dietary fibre	Obese and overweight pregnant women (BMI = 30.7 kg/m^2^)	Study began in early pregnancy (≤17 weeks)	— Modulated gut microbiota — Lowered maternal inflammatory status	[227]

^a^BMI, body mass index; ^b^LDL, low density lipoprotein; ^c^HDL, high density lipoprotein.

**Table 5 nutrients-11-01468-t005:** List of companies that offer gut microbiome testing.

Company (Website)	Method (as Indicated on Website)	Output for Consumer
Biohm (biohmhealth.com)	Sequences genes of bacteria and fungi at genus and species level	Consumer receives a grade of microbiome diversity, a comparison of all six major bacterial communities and four major fungal communities to normal levels and a strain by strain analysis of bacterial and fungal communities. Consumer receives personalized dietary, lifestyle and supplemental recommendations.
American Gut (americangut.org)	16s rRNA gene sequencing	A general overview of how the consumer’s microbial profile compares to other participants. A full list of microorganisms found in the sample and their relative abundances is provided. Note: American Gut is a crowd-funded microbiome research project and was started to provide a means to collect a large set of data surrounding the microbiome.
DayTwo (daytwo.com)	Sequences the DNA of microbiome. Consumer also provides blood test results including ^a^HbA1c	A scoring system rates thousands of different foods and food combinations based on the consumer’s biometrics, gut microbiome analysis, lifestyle factors and health questionnaire to yield a unique nutrition profile that enables blood-sugar balance. The consumer follows the scores to choose the foods which won’t increase blood sugar levels using the DayTwo personalised nutrition app.
THRYVE (thryveinside.com)	16s rRNA gene sequencing	Consumer receives a gut wellness score, gut diversity score, a ‘likelihood analysis’ of symptoms based on deficiencies in beneficial bacteria, as well as personalised dietary recommendations. Examples of symptoms: ‘More likely to feel beset by worries’; ‘More likely to feel fatigued and tired’; ‘More likely to have poor sleep’; ‘More likely to have itchy and dry skin’; ‘Difficulty maintaining a healthy weight’
UBIOME–Gut Explorer (ubiome.com)	Patented precision sequencing process	Gut Explorer: Consumer receives a comprehensive breakdown of the microbiome, it’s functioning, and how it compares to others. SmartGut: In conjunction with general practitioner (who orders the test), consumer receives a diversity score and breakdown of beneficial and pathogenic microorganisms associated with gut conditions like ^b^IBS, and ^c^IBD, including ulcerative colitis and Crohn’s disease, as well as microorganisms associated with metabolic conditions including obesity, diabetes and ^d^NAFLD.
Microba (microba.com)	Metagenomic sequencing	Consumer receives a detailed report describing gut microbiota diversity, members of microbial community including fungi and parasites, metabolic potential (protein, fat and carbohydrate breakdown, vitamin production), comparison with microbiome of others and personalised dietary recommendations.
AtlasBioMed (atlasbiomed.com)	^e^N/A	Consumer receives a grade of microbiome diversity, ability of gut to breakdown fibre, ratios of gut bacteria which influence disease including obesity, type II diabetes, heart disease and ^f^IBD, personalised food recommendations.
VIOME (viome.com)	Metatranscriptomics	All living gut microorganisms are analysed including bacteria, fungi, parasites, viruses, bacteriophages, archaea, yeast etc. Consumer receives a score of gut microbiome balance, pathway activities and functions, and their integrative impacts on metabolism, inflammation, and other wellness factors. Consumer receives personalized dietary, lifestyle and supplemental recommendations.
OME (ome.health)	^g^NGS	OME Heart Health and OME Weight Loss: Consumer receives score of bacterial diversity, full genetic bacterial profile, identification of beneficial and pathogenic bacteria. Also available to the consumer is a 12-week Heart Health or Weight Loss coaching programme with personalised meal plans and progress tracking etc.
BTS Ireland (btsireland.com)	N/A	Provides a selection of tests based on stool analysis. The Intestinal Colonisation test screens for the most beneficial and pathogenic bacteria.

^a^HbA1c, haemoglobin A1c; ^b^IBS, irritable bowel syndrome; ^c^IBD, inflammatory bowel disease; ^d^NAFLD,, non-alcoholic fatty liver disease; ^e^N/A, not available; ^f^IBD, inflammatory bowel disease; ^g^NGS, next-generation sequencing.

**Table 6 nutrients-11-01468-t006:** Proposed framework for scientific evidence assessment of microbiota-based dietary advice.

(i) Study Design and Quality	
Considerations: *Type of microbe/diet/outcome interaction* A relatively “simple” interaction with a single strain of bacteria, measuring the outcome (e.g., glucose response) over a number of weeks can give more confidence of “cause and effect.” A “complex” study may involve a prebiotic + several strains administered over several weeks or months to assess the weight management response and is likely to have higher inter-individual variation and therefore may be harder to establish cause/effect: is it the overall diet, or the microbes, or both, having an improved effect? The type of interaction also determines the confidence and the numbers of times a study should be repeated in order to have a level of confidence. However, there are pros and cons, and all types of studies are required. A simple or “direct” interaction gives confidence but the overall health benefit (e.g., short-term glucose) will be limited. A “complex” interaction is harder to give confidence but comes with a better overall health benefit (e.g., long-term weight management). *Levels of Interaction* A ‘direct’ interaction could be administration of a bacterial strain affecting glucose response. An intermediate interaction: specific prebiotics, fibre etc. with any type of response thus harder to determine if it is the nutrients or microbe growth, or both. An indirect interaction is the case where a mechanistic interaction between the microbe variant and the dietary component on a health biomarker, including disease, is affected to some extent but is also influenced by many other possibly unknown processes, and it may take years for symptoms to manifest. This type of interaction may not be fully explained physiologically or may be only demonstrated statistically.	
(ii) Biological Mechanism and Plausibility	
Considerations: Biological plausibility is a judgement based on the collected evidence of a microbe x ­diet interaction on a phenotype. An example of high biological plausibility could be a single microbial strain known to have benefits regarding saturated fat metabolism that leads to lower triglycerides and cholesterol. In this respect, Neville and colleagues recently proposed a variant of Koch’s postulates to provide a framework to establish causation in the case of a single strain in human microbiota research [269]. On the other hand, a vegan diet high in fibre affects the gut flora and over time the symptoms of metabolic syndrome improve—this type of interaction may not be fully explained physiologically or may only be demonstrated statistically.	
(iii) Probability Term	
Considerations: Assessing the validity of a putative microbe × diet interaction is generally complex, and as knowledge deepens, assessment of its validity will develop.	
Probability terms based on subjective probability range	
Probability term	Subjective probability range (%)
A. Convincing	> 90
B. Probable	66–90
C. Possible	33–66
D. Insufficient	< 33

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
