# Peer review of "Precision Nutrition and the Microbiome Part II: Potential Opportunities and Pathways to Commercialisation"

_nutrients, 2019, doi:10.3390/nu11071468_

Reviewer 1 Report

This is a comprehensive review which is Part II regarding the microbiome and precision medicine.

The review is well thought out and referenced. A few comments may be considered:

-page 5, discussion on probiotic intervention during pregnancy. It would be nice to have data added with pre/probiotics/synbiotics and effects on breast milk composition, breastfeeding and infant microbiome. 

-page 34, first paragraph - the sentences beginning with "For example, ...." are repeated in paragraph - delete one of them.

-It would be helpful to add more tables, figures to summarize each section and break up the text

Author Response

Comments and Suggestions for Authors.
This is a comprehensive review which is Part II regarding the microbiome and precision medicine. The review is well thought out and referenced. A few comments may be considered:
1. -page 5, discussion on probiotic intervention during pregnancy. It would be nice to have data added with pre/probiotics/synbiotics and effects on breast milk composition, breastfeeding and infant microbiome.
Response: The Reviewer has made a valid point and we have now added a section on the impact of probiotic, prebiotic and synbiotic administration to pregnant and lactating mothers on breast milk composition, the infant in terms of health and growth and the infant microbiome (lines 252-330).
2. -page 34, first paragraph - the sentences beginning with "For example, ...." are repeated in paragraph - delete one of them.
Response: We thank the Reviewer for noting this error and the extra paragraph has been deleted (line 1648).
3-It would be helpful to add more tables, figures to summarize each section and break up the text
Response: We agree with the Reviewer and as a consequence we have generated 4 new Tables for Sections 2 and 3 which summarize the main findings from the cited studies:
Section 2, Table 1: Potential Benefits of Probiotics, Prebiotics, Synbiotics and Fibre on Different Life Stages
Section 3, Table 2: Impact of Probiotics on Overnutrition in Animal Studies and Human Intervention Studies
Section 3, Table 3: Impact of Prebiotics and Synbiotics on Overnutrition and Undernutrition in Animal Studies and Human Intervention Studies
Section 3, Table 4: Impact of Fibre on Overnutrition in Animal Studies and Human Intervention Studies

Reviewer 2 Report

This is a very well researched review of a large field, and has resulted in a very long paper.

Much of parts 2 and 3 are a recital of many trials and what was found. It would be good (but may not be possible) to create a table for each section, summarizing these trials this is just a suggestion, not a requirement, but it would make it more readable.

Page 34 seems to have a lot of repetition about the F. prausnitzii subgroups and I think could be made more concise.

There are numerous small grammatical errors, use of wrong word and missing commas. I have highlighted these in a copy that I am unable to upload here, but will send separately to the editors. Also there are a couple of questions I have inserted.

I note the use of 3 letter abbreviations for genera following first use. Please check this is consistent with the guidelines for this journal.

Author Response

Comments and Suggestions for Authors
This is a very well researched review of a large field, and has resulted in a very long paper.

1. Much of parts 2 and 3 are a recital of many trials and what was found. It would be good (but may not be possible) to create a table for each section, summarizing these trials this is just a suggestion, not a requirement, but it would make it more readable.
Response: We agree with the Reviewer and as described in Response 3 of Reviewer 1, we have added 4 new Tables to the review which summarize the information discussed in Sections 2 and 3.
2. Page 34 seems to have a lot of repetition about the F. prausnitzii subgroups and I think could be made more concise.
Response: We thank the Reviewer for noting this error and the extra paragraph has now been deleted (line 1648).
3. There are numerous small grammatical errors, use of the wrong word and missing commas. I have highlighted these in a copy that I am unable to upload here, but will send separately to the editors. Also there are a couple of questions I have inserted.
Response: We have carefully read the review again and made corrections.
4. I note the use of 3 letter abbreviations for genera following first use. Please check this is consistent with the guidelines for this journal.
Response: We have now correctly abbreviated names of all genera throughout the review. For Bifidobacterium, the abbreviation Bif. is used and for Bacteriodes the abbreviation Bac. is used to avoid confusion between the two. This approach was also used in Part 1 of this review which has been published in Nutrients.